# Histone lactylation: A new epigenetic mark in the malaria parasite *Plasmodium*

Ibtissam Jabre[1], Nana Efua Andoh[1], Juliana Naldoni[1], William Gregory[1],
Haddijatou Mbye[1], Chae Eun Yoon[2,3], Aubrey J. Cunnington[2,3], Athina Georgiadou[2,3],
Andrew M. Blagborough[1], Catherine J. Merrick[1]*

1 Department of Pathology, University of Cambridge, Cambridge, United Kingdom, 2 Section of Paediatric Infectious Disease, Department of Infectious Disease, Imperial College London, London, United Kingdom, 3 Centre for Paediatrics and Child Health, Imperial College London, London, United Kingdom

☯ These authors contributed equally to this work.
* cjm48@cam.ac.uk

## Abstract

Epigenetic processes play important roles in the biology of the malaria parasite *Plasmodium falciparum*. Here, we characterised a new epigenetic mark, histone lactylation, for the first time in *Plasmodium*: it was found in two human malaria parasites, *P. falciparum* and *P. knowlesi*, and also *in vivo* in two rodent malaria models, *P. yoelii* and *P. berghei*. Histones were increasingly lactylated in response to elevated lactate levels *in vitro* and *in vivo*, making this mark uniquely well-placed to act as a metabolic sensor, since severe falciparum malaria characteristically leads to hyperlactataemia in the human host. Mass spectrometry showed that lysines on several parasite histones could be lactylated, as well as many non-histone chromatin proteins. Histone lactylation was less abundant and less inducible in *P. knowlesi* than *P. falciparum*, suggesting that *P. falciparum* may have evolved particular epigenetic responses to this characteristic feature of its pathology. Finally, in the rodent model *P. yoelii*, hyperlactataemia correlated with parasite transcriptomic programmes that suggested metabolic 'dormancy'.

## Author summary

The pathology of human malaria often includes 'hyperlactataemia': blood lactate rising to aberrantly high levels. This is also seen, for example, in sepsis. It predicts severe and fatal malaria because it can cause respiratory distress in the patient. During malaria, lactate is generated by the parasite's own metabolism and also by hypoxic host tissues. For the parasite, it could be advantageous to sense this state of pathology and respond by modulating virulence. Here, we report that this could occur epigenetically, with parasite histones and other proteins being flexibly marked with lactyl-lysine when lactate levels are high. This

**Data availability statement:** Sequencing data from mouse, P.yoelii and P. berghei are available at ENA under the ID PRJEB43641. Proteome data are deposited at the ProteomeXchange Consortium via the PRIDE partner with dataset identifier PXD055236. All other relevant data are in the manuscript and its Supporting information file.

**Funding:** This work was funded by Wellcome Discovery Award 225171/Z/22/Z to CJM. AG is supported by an Imperial College Research Fellowship. AMB is funded by the MRC [MR/N00227×/1 and MR/W025701/1], Sir Isaac Newton Trust, Alboroda Fund, Wellcome Trust ISSF and University of Cambridge JRG Scheme, GHIT, Rosetrees Trust (G109130) and the Royal Society (RGS/R1/201,293 and IEC/R3/19,302). The funders had no role in study design, data collection and analysis, decision to publish, or preparation of the manuscript.

**Competing interests:** The authors have declared that no competing interests exist.

is an entirely novel aspect of chromatin biology in malaria parasites. We found it in several parasite species from both humans and rodents, but it occurred particularly in those that characteristically cause hyperlactataemia, suggesting the evolution of adaptive host/parasite sensing pathways. Indeed, in one such rodent model, parasite genes that were heavily lactylated did show modulated expression in high-lactate hosts, strongly suggesting a direct epigenetic pathway. Overall, gene expression patterns in such hosts suggested a parasite 'dormancy' response, which could promote parasite survival during severe pathology. Overall, lysine lactylation in malaria parasites is exceptionally well-placed to act as a metabolic sensor to modulate virulence.

## Introduction

Human malaria occurs when protozoan *Plasmodium* parasites infect red blood cells. Six *Plasmodium* species can cause malaria in humans, including the main agent of severe and fatal malaria, *P. falciparum*, and the zoonotic macaque parasite *P. knowlesi.* These two are currently the only human malaria parasites that can be cultured continuously *in vitro.*

Clinical malaria has characteristic pathological features, one of which is hyperlactataemia. Hyperlactataemia is also seen, for example, in sepsis, and is clinically defined as a level of blood lactate above 5mM. Patients with falciparum malaria can reach at least 15mM [1,2]. Such high levels of blood lactate can come from several sources: parasites in the bloodstream respire by glycolysis, producing lactate, and they also cause their host erythrocytes to adhere to capillary walls, impeding blood flow and promoting anaerobic respiration in host tissues. Collectively, this (along with other more complex pathologies) can result in hyperlactataemia, which can lead to the fatal malaria syndrome of metabolic acidosis and respiratory distress. Thus, malarial hyperlactataemia is predictive of severe and fatal disease [3].

*Plasmodium* parasites could benefit from being able to sense the metabolic environment in their host. By doing so, they could modify their virulence accordingly – for example, by increasing commitment to sexual stages that are competent for transmission to new hosts, or by modulating cytoadherence of infected erythrocytes. This fascinating area of host-parasite biology is increasingly being explored, with many aspects of blood chemistry, including levels of glucose, magnesium and S-adenosylmethionine, being proposed to influence the virulence of *P. falciparum* parasites [4–7]. In some cases, such as that of S-adenosylmethionine, the blood metabolite influences parasite virulence through epigenetics, because the virulence phenotype concerned – sexual stage commitment or 'gametocytogenesis' – is controlled by an epigenetic switch.

Blood lactate has yet to be fully explored in this context, but since it correlates with parasite load and disease severity, it could potentially serve as both a 'quorum sensor' and a measure of host stress. In a human infection study, we previously linked lactate to the expression pattern of a key virulence gene family, *var*, which encodes

variant adhesins expressed on infected erythrocytes [1]. Specifically, when patients had high blood lactate, their malaria parasites expressed high levels of virulence-associated *var* genes. The *var* gene family is known to be epigenetically controlled by histone acetylation and methylation.

In fact, epigenetics plays many prominent roles in *Plasmodium* biology besides cytoadhesion. It controls aspects of virulence including alternate invasion pathways and sexual stage commitment (gametocytogenesis), as noted above [8]. In a recent *in vitro* study, gametocytogenesis was also linked to lactate, with parasites cultured in added lactate converting to gametocytes at an elevated rate [9].

Here, we explored lactate as a potential epigenetic modifier in *Plasmodium*, following the recent discovery of lactyl-lysine as a new epigenetic mark in mammalian cells, similar to the well-characterised acetyl-lysine [10] (Fig 1A). Since its discovery in 2019, this epigenetic mark has been heavily explored in the context of human cells in high-lactate environments – primarily tumours experiencing the Warburg effect [11,12]. It has not yet been explored in the context of malaria, although this is a disease characterised by systemic hyperlactataemia. However, we did recently observe that histone lactylation could occur in *P. falciparum*, and hence proposed this as a new virulence modifier in *Plasmodium* [13]. We now report that the epigenetic mark is conserved in several *Plasmodium* species, occurring both *in vitro* and *in vivo*. It appears on many histone lysine residues and is responsive to lactate levels both in culture and in mammalian host blood. Therefore, it could have many virulence-modulating roles.

## Results

### *Plasmodium* histones are inducibly lactylated

We first used western blotting to measure the level of lactyl-lysine in *Plasmodium* histones after treating cultured parasites with increasing levels of lactate. Trophozoite-stage *P. falciparum* parasites were exposed to 5-25mM lactate, added to the culture media for 12h. Fig 1B shows that their histones were inducibly and titratably lactylated, with histone H4 showing the strongest lactyl-lysine signal. An antibody identifying a single lactylated residue on histone H4 (K12) confirmed that the general lactyl-lysine antibody recognised this histone, but lactylation of the K12 residue itself was not inducible (Fig 1C). We conducted the same experiments on a second species, *P. knowlesi*, and showed that histone lactylation was similarly inducible (Fig 1D). It was, however, much less abundant than in *P. falciparum*, since greater amounts of parasite extract were required to detect comparable signals on *P. knowlesi* histones, and the specific residue H4K12La was barely detectable (Fig 1E).

We then investigated whether this phenomenon occurs naturally *in vivo* during malaria. The rodent model species *P. yoelii*, strain 17XL, can generate severe hyperlactataemia [14], and blood lactate levels in mice infected with *P. yoelii* 17XL did indeed correlate with histone lactylation in the parasite (Fig 1F). Notably, not all rodent malaria models lead consistently to hyperlactataemia as infection progresses, and in the *P. berghei* ANKA strain, which does not [14], we did not observe a strong correlation between blood lactate and histone lactylation (S1 Fig). Overall, these results revealed that there is a conserved pathway for histone lactylation in *Plasmodium*, induced by hyperlactataemia and apparently most prominent in species that do characteristically cause hyperlactataemia.

### Histone lactylation varies across the *P. falciparum* cell cycle

We next investigated the cell-cycle dynamics of the lactyl modification, again using western blotting. Fig 1B–E focuses only within the trophozoite stage, showing that lactylation is induced within 12h and 8h in *P. falciparum* and *P. knowlesi* trophozoites respectively. Fig 2A shows that there is a natural cell-cycle-related pattern in histone lactylation, probably because parasite metabolism, which peaks in trophozoites, produces its own endogenous lactate. Histone lactylation peaked in the trophozoite stage in two successive parasite cycles and fell in ring and schizont stages.

To measure the extent of endogenous lactate production in our parasite culture system, we measured lactate levels in the media using a biochemical assay, both when parasites were cultured under standard conditions (reducing the

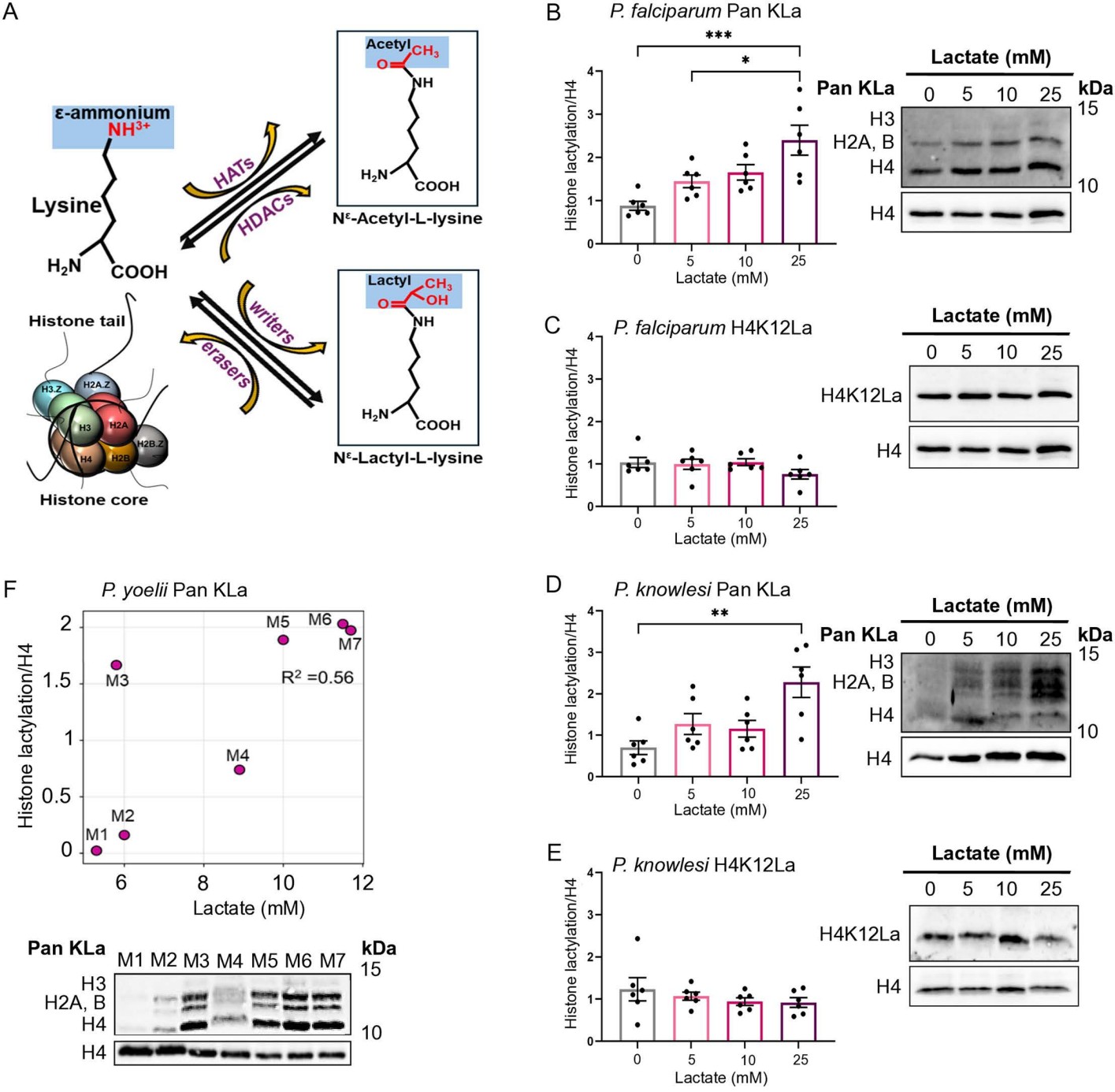

**Fig 1. *Plasmodium* histones are inducibly lactylated.** A: Schematic showing that lysine residues on histone tails can be either acetylated (by histone acetyltransferases (HATs), with deacetylation via histone deacetylases (HDACs)), or lactylated by writers and erasers that are not yet comprehensively identified. B, C: Lysine lactylation was measured in *P. falciparum* parasites grown in different concentrations of L-lactate for 12h. Western blots show 'Pan KLa' (B) and histone H4 K12 lactylation ('H4K12La' **(C)**), as well as total histone H4 as a control. Graphs show quantifications of blots from n = 6 independent experiments, with each lactyl-histone signal quantified as a proportion of the total histone H4 signal (bars show means, dots show individual values, error bars show SEM). Multiple comparison of means was performed using one-way ANOVA and Tukey HSD post hoc test: *, $p < 0.05$; ***, $p < 0.001$; comparisons not shown, not significant. A representative western blot from each group of 6 experiments is shown. D, E: Lysine lactylation was measured in *P. knowlesi* parasites grown in different concentrations of L-lactate for 8h, as in **(B, C)**. F: Lysine lactylation was measured in *P. yoelii* 17XL

parasites exposed to varying levels of lactate in the blood of the host mouse. Western blots show 'Pan KLa' and histone H4 as control. M1-M7, n = 7 individual mice. Scatter plot represents Pearson correlation (R² = 0.56, p = 0.05) between histone KLa signal intensities normalized to H4 (y-axis) and lactate levels in the blood (x-axis) upon parasite collection from the host.

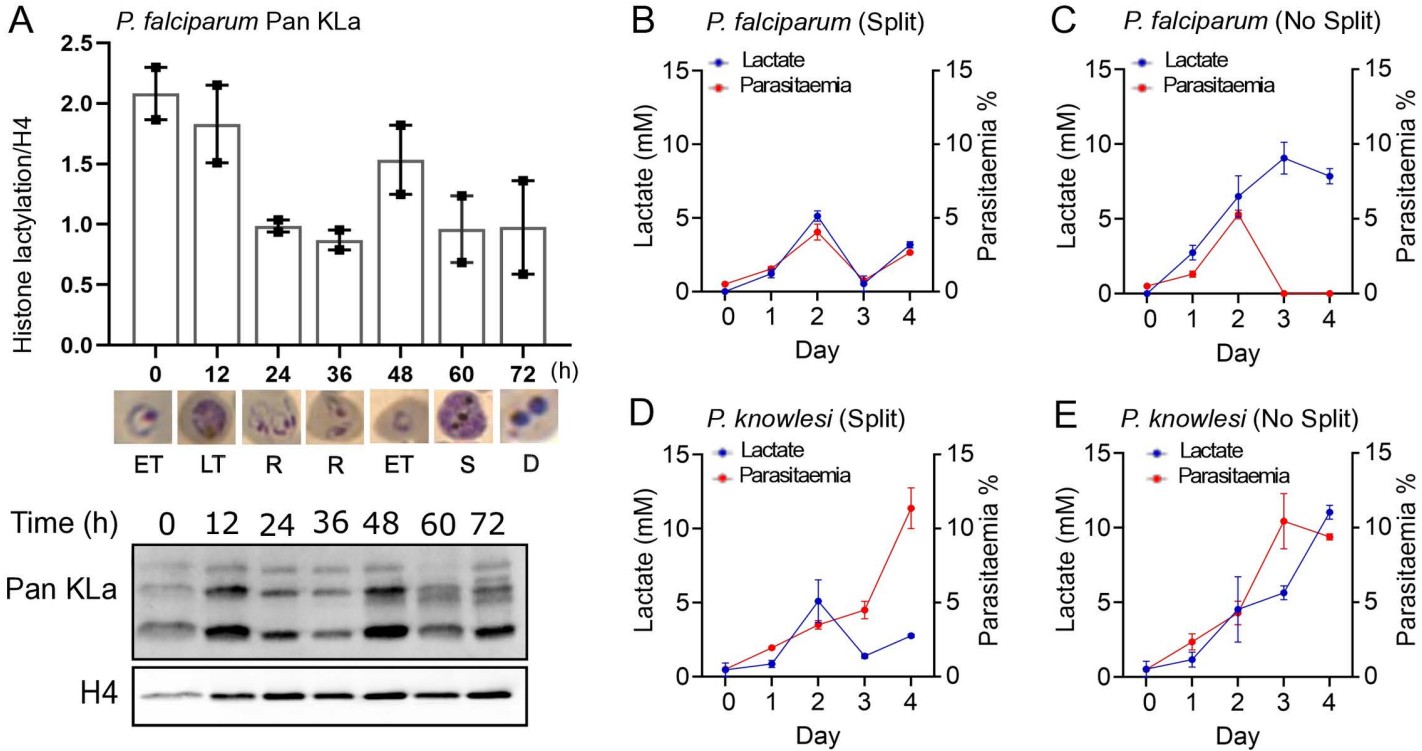

**Fig 2. Histone lactylation varies across the *P. falciparum* cell cycle.** A: *P. falciparum* parasites were grown from 1% parasitaemia without splitting or changing the media for 72h, starting at the early-trophozoite stage and sampling every 12h. Stage of the parasite culture is shown on the x-axis, and example parasite morphology is also shown. By 60h, morphology showed that the culture was starting to crash, with dead (D) pyknotic parasites by 72h. The graph shows the intensity of each KLa signal relative to total H4, quantified from n = 2 biological experiments; one representative western blot is shown. B-E: Lactate levels and parasitaemias were measured every 24h in cultures of *P. falciparum* (B, C) and *P. knowlesi* **(D, E)**, starting at 0.5% parasitaemia, either split back to 0.5% every 48h (B, D) or allowed to grow continuously without splitting **(C, E)**. Data shown are the mean of n = 2 biological experiments.

parasitaemia and changing the media every 48h), and also when they were allowed to grow continuously to the point of 'crashing' (the term for parasite death caused by high parasitaemia and exposure to spent media [15]). As expected, media lactate levels correlated with parasitaemia, reaching only ~5mM within 2 days as a culture of *P. falciparum* grew from 0.5% to 4.6% parasitaemia (Fig 2B). When the culture was allowed to 'crash', lactate levels peaked at ~10mM (Fig 2C). The profile of lactate production was similar in *P. knowlesi* cultures (Fig 2D), although this parasite is more resistant than *P. falciparum* to 'crashing', so these parasites could grow exponentially for at least 3 days, producing lactate continuously and reaching 11.5mM by day-4 (Fig 2E).

Overall, these data show that routine culture of both parasite species can expose them to significant but modest lactate fluctuations. Routine cultures can reach the clinical threshold for hyperlactataemia (5mM), which is sufficient to induce some histone lactylation (Fig 1B and 1D), but is much lower than the levels attained in human patients with severe malaria [1,2].

## Histone acetylation is not strongly affected by exposure to high lactate

In human cells, both acetylation and lactylation have been detected on the same histone lysine residues, and it has been suggested that histone acetyltransferase enzymes may moonlight as lactylases [10]. This raised the possibility that in *Plasmodium*, histone lactylation could reciprocally affect histone acetylation – a well-established factor in gene expression regulation.

We tested this by western-blotting for acetyl H4 and acetyl H3 under conditions of high lactate exposure, as in Fig 1. Fig 3A–D shows no statistically significant change in acetylation of H3 or H4, despite a trend towards lactate-inducible acetylation. Thus, acetylation was not strongly lactate-inducible, although it is possible that the two histone marks may be somewhat mutually regulated by lactate.

## Histone lactylation is distributed throughout the nucleus

In *P. falciparum,* certain histone modications are specifically associated with heterochromatin, which tends to be located at the nuclear periphery, marked by heterochromatin protein 1 (HP1) [16], whereas other modifications associated with euchromatin appear throughout the nucleoplasm. We investigated the location of histone lactylation using immunofluorescence microscopy (Fig 4). We found no evidence for compartmentalisation of lactylated histones, with both the general lactyl-lysine signal (Fig 4A) and the specific signal for H4K12 (Fig 4B) appearing throughout the nuclei of all parasite

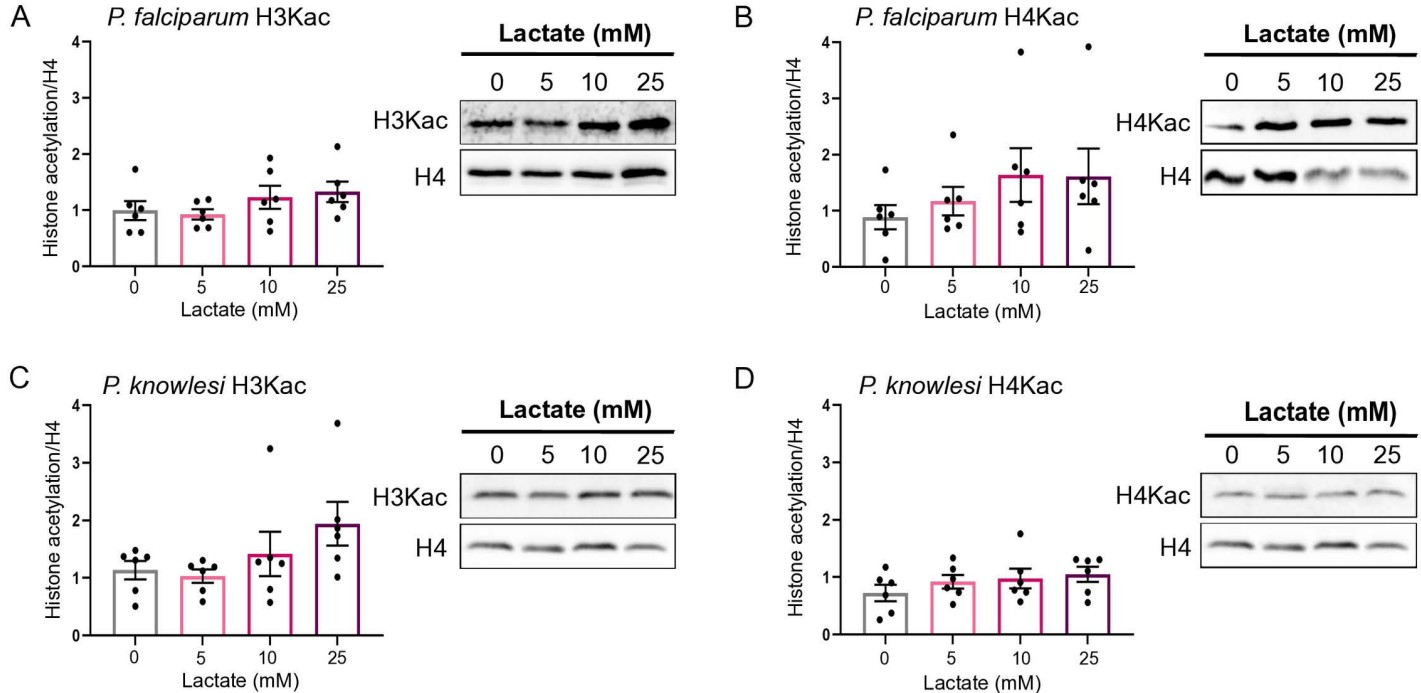

**Fig 3. Histone acetylation is not strongly affected by exposure to high lactate.** A,B: Histone 3 lysine acetylation ('H3Kac', **(A)**) and histone 4 lysine acetylation ('H4Kac', **(B)**) were measured in *P. falciparum* parasites grown in different concentrations of L-lactate for 12h, as in Fig 1. Graphs show quantification of blots from n = 6 independent experiments, with each acetyl-histone signal quantified relative to histone H4 (bars show means, dots show individual values, error bars show SEM). Multiple comparison of means was performed using one-way ANOVA and Tukey HSD post hoc test: none were significantly different at $p < 0.05$ (lowest p-values 0.35 for H3Kac, 0.52 for H4Kac). A representative western blot from each group of experiments is shown. C,D: Histone 3 lysine acetylation ('H3Kac', **(C)**) and histone 4 lysine acetylation ('H4Kac', **(D)**) were measured in *P. knowlesi* parasites grown in different concentrations of L-lactate for 8h, as in Fig 1. Data represented and analysed as in **(A, B)**; no trends were significantly different at $p < 0.05$ (lowest p-values 0.15 for H3Kac, 0.40 for H4Kac). A representative western blot from each group of experiments is shown.

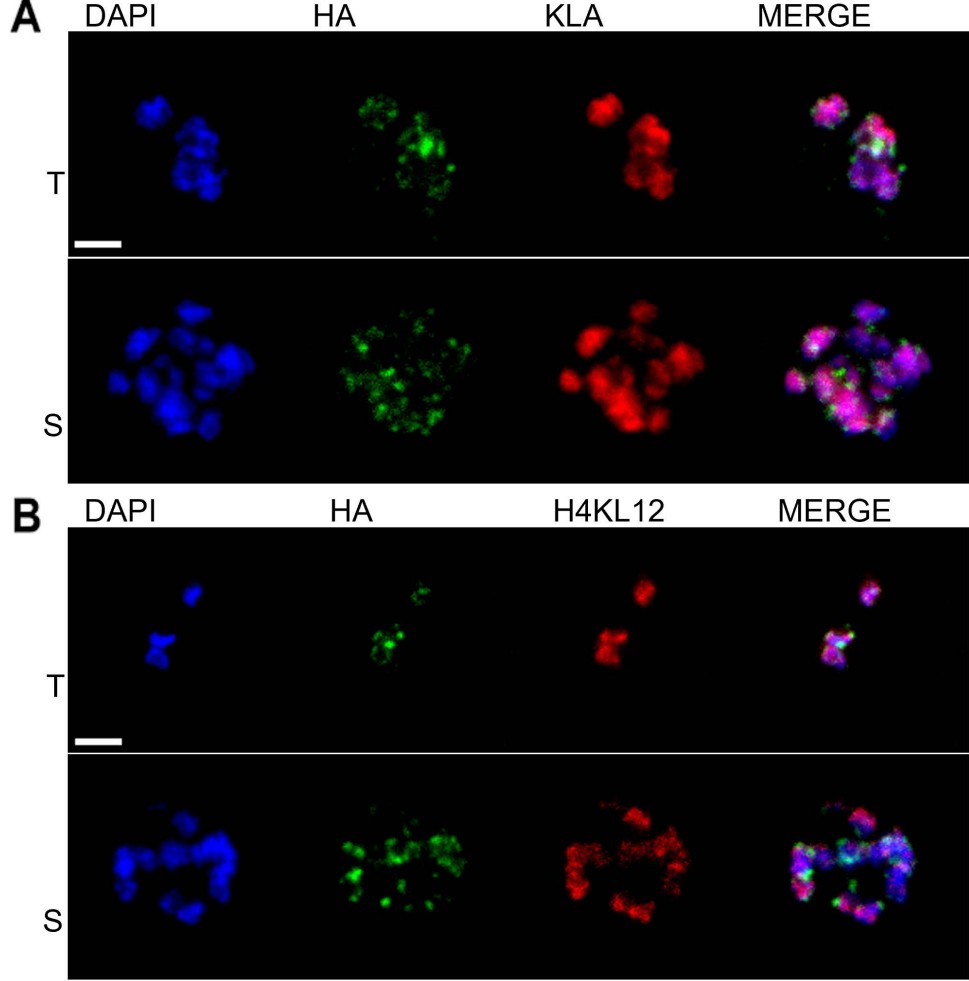

**Fig 4. Histone lactylation is distributed throughout the nucleus.** Confocal microscopy images showing the location of lactyl PTMs and of HP1 in *P. falciparum* 3D7 HP1_HA (this line expresses heterochromatin protein 1 with a haemagglutinin (HA) tag). Lactyl-lysine, red; HP1_HA, green; DAPI, blue. KLa (A) and H4K12La (B) distributions are mostly nuclear, and do not colocalise specifically with HA-tagged HP1. T, trophozoite stage; S, schizont stage. Scale bar (2 μm) applies to all images.

stages. Neither signal colocalised with HP1, and both appeared similar to the pan-nuclear distribution of histone acetylation (S2 Fig). The great majority of lactyl-lysine signal was nuclear (Fig 4A), suggesting that there is little lactylated protein in the cytoplasm of *Plasmodium.*

## Histone lactylation occurs on multiple histone lysine residues

For a comprehensive view of the histone lysine residues that can be lactylated in both *P. falciparum* and *P. knowlesi*, we used liquid chromatography-tandem mass spectrometry (LC-MS/MS). Trophozoite-stage parasites were either exposed or not exposed to 25mM lactate and their histones were acid-extracted from chromatin in biological triplicate. In a single pilot experiment, histones were also extracted from mixed stage cultures (+/- 25mM lactate), enabling detection of KLa sites in all stages of the asexual cycle. Histone integrity and the induction of lactylation were confirmed via Coomassie-stained SDS-PAGE and by immunoblotting for KLa (S3 Fig). (25mM lactate, for 12h in *P. falciparum* or 8h in *P knowlesi* which has

a shorter cell cycle, was chosen for these and all subsequent experiments. From Fig 1, this induced the strongest histone lactylation, without markedly affecting parasite growth (S4 Fig), whereas even higher levels of ≥ 30mM over 48h were previously reported to affect parasite growth in *P. falciparum* [17]).

Mass spectrometry on the histone-containing region of the gel identified all eight histones (H2A, H2A.Z, H2B, H2B.Z, H3, H3.3, CenH3 and H4), with sequence coverage >67.5% except in the case of CenH3 (S1–S4 Tables). We then mapped KLa sites, as previously reported [10] (S5A Fig), revealing 16 and 21 KLa sites in *P. falciparum* and *P. knowlesi* respectively (Fig 5A). These were in *P. falciparum*: H4 K5, 8, 12, 67; H3 K14, 27, 37, 38; H3.3 K27; H2B K49; H2A.Z K18, 29; and H2B.Z K8, 14, 18, 53. In *P. knowlesi*, only a subset of the sites were the same: H4 K8, 12, 31, 67; H3 K18, 23; H2B K35, 38, 49, 64, 100, 108; H2A.Z K14, 29; and H2B.Z K8, 14, 29. They were unevenly distributed across the histone tails (for example, none on H2A versus four on H4), and almost all KLa sites could also be acetylated (S5B Fig and S1–S4 Tables). Interestingly, despite the histone sequences being highly conserved between *Plasmodium* species, the majority of KLa sites were species-specific, suggesting that this modification may have species-specific function(s). As expected from Fig 1, lactylation of some residues was induced after exposure to 25mM lactate, but inducibility varied between residues and also between species (Fig 5B). It also varied in trophozoites versus mixed-stage cultures, implying that some residues may be lactylated stage-specifically.

### Cross-talk may occur between histone lactylation, acetylation and methylation

Acetylation and lactylation were often detected on the same lysine residues, both in our data and previously in human cells [10]. (Since the same lysine linkage is used (Fig 1A), the same site on the same histone molecule cannot be simultaneously modified with both, but histones within a population of molecules can be modified with either.) Therefore we sought to investigate whether lactate exposure in the trophozoite stage could induce changes in the distribution of histone acetylation (Kac), as well as di- and tri- methylation (K(me2), K(me3)).

There was no dramatic change in the distribution of Kac sites after exposure to 25mM lactate, and Kac remained the most abundant histone modification, as previously reported [18] (S1–S4 Tables). Almost all KLa sites were also detected as Kac but the converse was not true: ~65% of Kac sites were unique (S5B Fig). After lactate exposure, 44% of histone KLa sites in *P. falciparum* and 17% of those in *P. knowlesi* were significantly induced (*P*-value ≤ 0.05) (Fig 6A, top panel and S1–S2 Tables), corroborating the observation from Fig 1 that lactylation was more strongly inducible in *P. falciparum* than in *P. knowlesi*. By contrast, only ~12% and ~7% of histone Kac sites, respectively, were significantly up- or down-regulated. Hence, relatively few Kac sites (amongst very many that can be acetylated) must be responsible for the modest rise in total histone acetylation after lactate exposure, which was detected by western blot in Fig 3. Overall, KLa was clearly the most inducible post-translational modification (PTM) after lactate exposure (S6A Fig), and sites that were strongly induced as lactylated were not simultaneously induced as acetylated (S6B Fig).

In contrast to Kac, major changes did occur in K(me2) and K(me3) after lactate exposure: ~90% of Kme sites were significantly down-regulated in *P. falciparum* (Fig 6A, lower panel and S1–S2 Tables). This was not the case in *P. knowlesi*, where no reciprocal changes in K(me2) and K(me3) were seen.

### Non-histone chromatin-associated proteins can be lactylated

Chromatin-associated proteins besides histones appeared in the mass spectrometry data, and many of these were also lactylated, with a subset of the modifications being induced after exogenous lactate exposure (Fig 6B and S3–S4 Tables). In *P. falciparum*, lactylated sites were found in chromatin-bound proteins such as Alba3, several RNA helicases, proteins implicated in epigenetics (e.g., histone deacetylase 2), in glycolytic metabolism (e.g., glycerol-3-phosphate dehydrogenase) and in transcriptional control (e.g., two ApiAP2 transcription factors). A third AP2 factor, AP2G, which induces gametocytogenesis, was also detected as lactylated, but its coverage fell below the threshold for inclusion, probably due to its very low expression in these asexual-stage cultures. An acetyl-CoA transporter was also lactylated, suggesting again

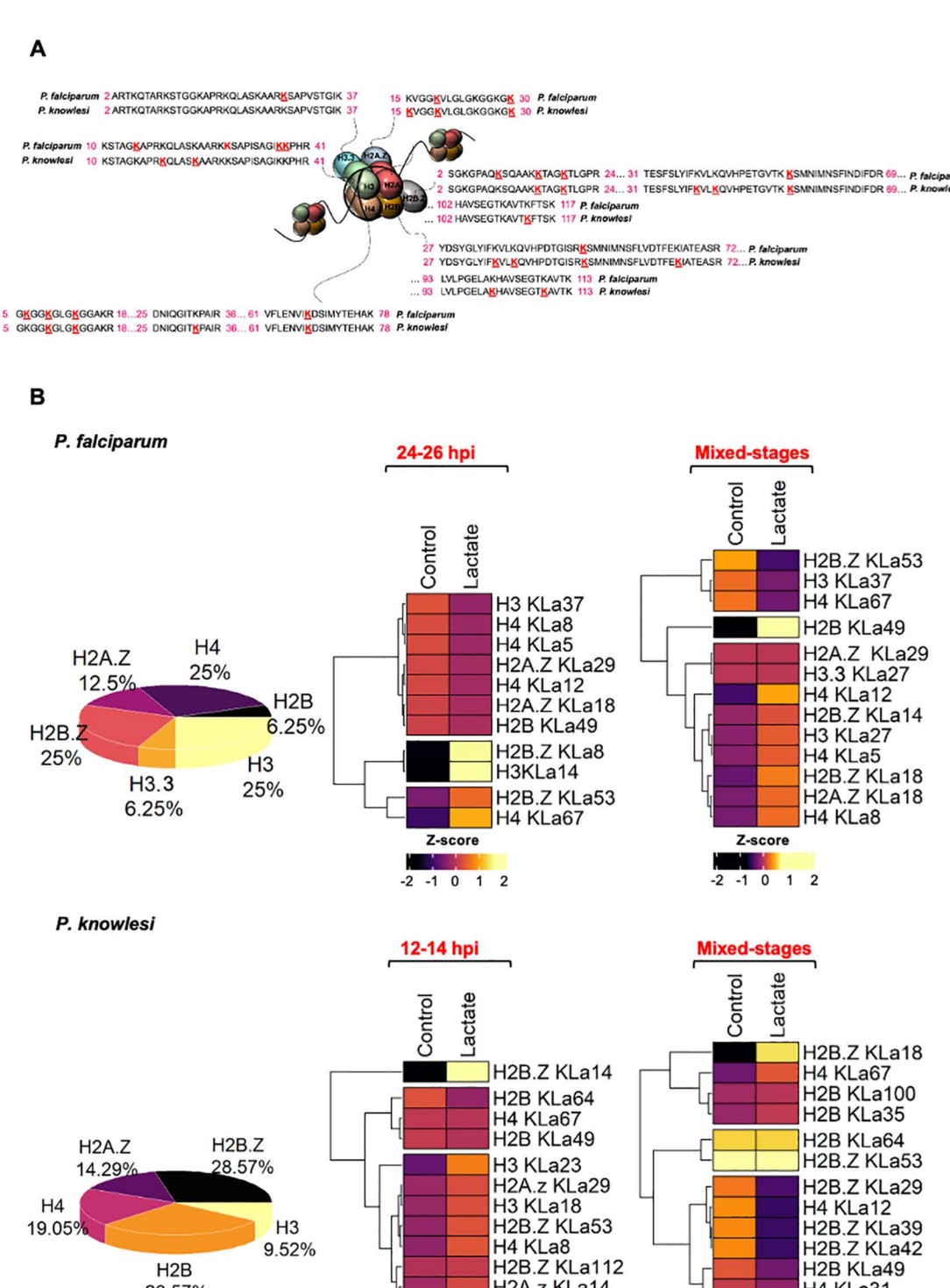

**Fig 5. Histone lactylation occurs on multiple histone lysine residues in two *Plasmodium* species.** A: Sequence alignments of histone tails in *P. falciparum* and *P. knowlesi*, showing in red all KLa sites detected by mass spectrometry (either in trophozoite or mixed-stage cultures, +/- 25mM lactate). Numbers in pink flanking the peptide sequences are residue numbers of the N- and C-termini of the peptides. The tail of histone H2A is not shown

because no KLa was detected on this histone. B: Pie charts representing the distribution of KLa sites from **(A)**, and heatmaps representing normalized intensities of KLa-containing peptides (Z-score transformed). Hierarchical clustering was used to partition the KLa sites into 3 clusters with euclidean distance and ward.D clustering algorithm. The time windows for trophozoite cultures of *P. falciparum* and *P. knowlesi* were 24-26 and 12-14 hpi, respectively, and these were treated with lactate for the subsequent 12 or 8h.

the possibility of cross-talk between acetyl and lactyl epigenetic pathways (S7A Fig and S3 Table). In *P. knowlesi*, most of the same proteins were not detected as lactylated, although other proteins, including the acetyltransferase GCN5, were lactylated (S7B Fig and S4 Table).

Although almost all the lactylated lysines within histones could also be acetylated (S5B Fig), this was not the case in non-histone proteins. Unique, lactate-induced KLa sites appeared on several chromatin-associated proteins. These included epigenetic regulators (e.g., SAP18, associated with HDAC1) and RNA binding proteins such as Alba4 in the case of *P. knowlesi*, and RNA helicases (e.g., NAM7, responsible for RNA turnover), RNA binding proteins, AP2 transcription factors and gametocytogenesis factors (e.g., Male Development 1) in the case of *P. falciparum* (S1–S2 Tables).

Collectively, it appeared that the KLa modification was widespread, both on histones and on other proteins. It was modulated on some proteins by exogenous lactate, but also probably derived from lactate produced by the parasite. The wide range of lactylated proteins suggests that the modification could regulate multiple aspects of *Plasmodium* biology, from epigenetics to RNA biology to protein function.

### Lactyl post-translational modification does not correlate with levels of protein expression

There was no consistent correlation between proteins that were inducibly modified and proteins that changed in abundance after exposure to 25mM lactate (S8 Fig). In *P. knowlesi*, no histones and very few other proteins showed significant changes (with a sca.$P$-value ≤ 0.01) after lactate exposure; in *P. falciparum*, a few proteins did change significantly in abundance, including histone H2A.Z and HP1, and interestingly the changes were mostly downregulation (S8A Fig). Overall, however, there was no correlation between proteins whose PTMs were significantly changed, and proteins that changed in abundance (S8B Fig). Therefore, protein lactylation does not appear to be a consistently 'stabilizing' nor 'destabilizing' PTM.

### Proteins in different biological pathways are lactylated in *P. falciparum* versus *P. knowlesi*

It was apparent that the PTMs affected by lactate exposure, both on histones and on non-histone proteins, differed in *P. falciparum* versus *P. knowlesi.* Gene ontology (GO) enrichment analysis was therefore performed to gain insight into the pathways most affected in each species.

Amongst the proteins whose PTMs changed after lactate exposure, many different biological functions were found to be enriched *(P*-value ≤ 0.05), depending on both the species and the epigenetic modification (KLa, Kac or Kme). In *P. falciparum,* lactate-modulated KLa sites were significantly enriched in proteins involved with metabolism of tRNAs and indole compounds, for example; in *P. knowlesi,* the top GO terms associated with such KLa sites included 'cell growth', 'cell division' and 'autophagy' (Figs 7A and S9 and S5 Table).

Amongst GO terms for cellular components, KLa sites, as well as Kac and Kme sites, that responded to lactate exposure were enriched in proteins involved with the nucleosome and protein-DNA complexes. This was not surprising given the known predominance of such PTMs on histones, as well as the location of KLa observed in Fig 4, which was almost exclusively nuclear. In *P. knowlesi*, there was also an interesting enrichment of lactate-modulated KLa sites in the carbon CCR4/Not complex, which regulates aspects of protein function, RNA metabolism and transcriptional activity during late gametocyte differentiation and post-fertilization development [19] (Figs 7B and S10 and S5 Table). Amongst molecular function GO terms, both species showed enrichment of 'protein dimerization' and 'catalytic activity', amongst others (S11 Fig and S5 Table).

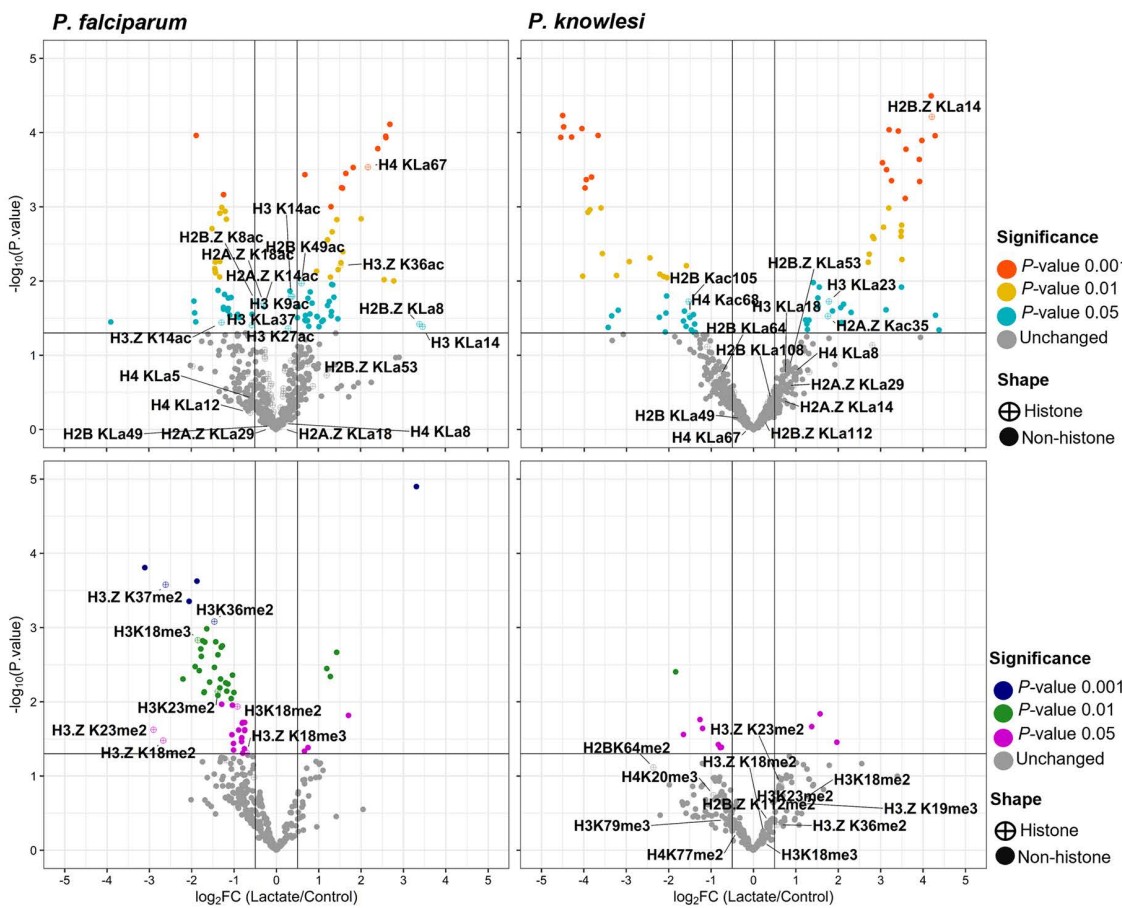

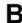

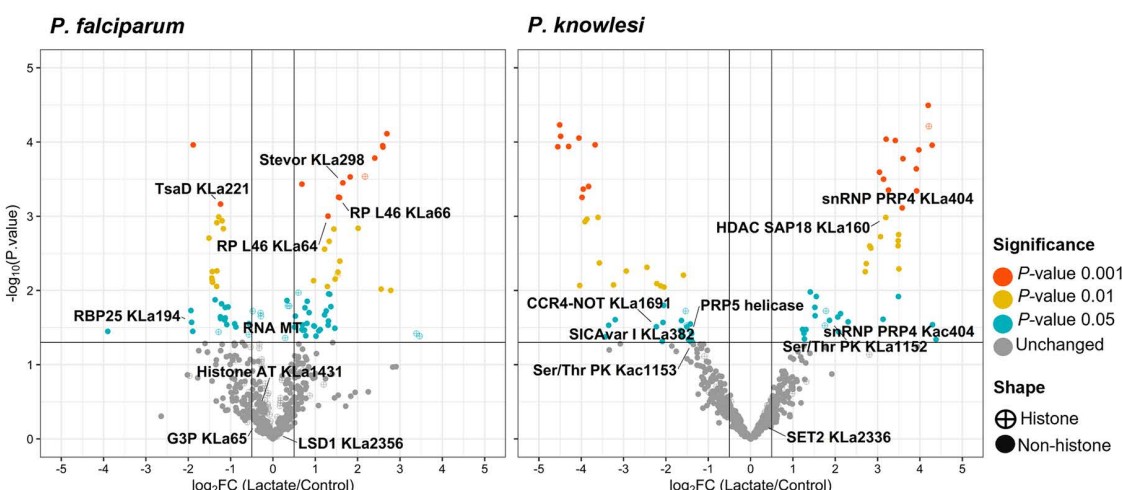

**Fig 6. Changes in PTMs on both histones and non-histone proteins are induced by exposure to high lactate.** Volcano plots representing (A) histones with differential lysine lactylation (KLa) and acetylation (Kac), in the top panels, and differential tri- or di-methylation (Kme3, Kme2) in the lower panels; (B) non-histone proteins with differential lysine lactylation (KLa). In all plots, data are from trophozoite cultures of *P. falciparum* or *P.*

*knowlesi*, +/- 25mM lactate treatment. X-axes represent $\log_2$ fold changes (FC) calculated from normalized intensities, and y-axes represent *P*-values. Colored dots represent PTMs that changed significantly after lactate exposure, based on their significance threshold indicated by *P*-value. (Grey dots represent those that did not change significantly.) Symbols for histones and non-histones are hatched and filled circles, respectively. Horizontal and vertical lines represent *P*-value and $\log_2$FC significance thresholds. In **(B)**, gene name abbreviations in *P. falciparum*, on the left: RNA MT, RNA methyltransferase; Histone AT, Histone acetyltransferase; LSD1, lysine specific histone demethylase; TsaD, TSaD-domain-containing protein; RBP25, RNA binding protein 25; RP L46, Ribosomal protein L46; G3P, glyceradehyde-3-phosphate dehydrogenase. In *P. knowlesi* on the right: snRNP PRP4, small nuclear ribonucleoprotein; HDAC SAP18, histone deacetylase subunit SAP18; Ser/Thr PK, serine/threonine protein kinase, CCR4-NOT, CCR4-NOT complex; PRP5 helicase, Pre-mRNA-processing ATP-dependent RNA helicase; SET2, SET2 histone methyltransferase.

## Blood lactate levels in malaria-infected mice correlate with distinct transcriptomic profiles in the causative parasites

The foregoing data were all derived from parasites in *in vitro* culture: an environment that may not full mimic the hyperlactataemic host. However, in Fig 1F, we investigated a rodent malaria model that becomes severely hyperlactaemic, *P. yoelii* 17XL, and showed that histone lactylation in the *P. yoelii* parasites was strongly induced in hyperlactataemic mice. Rodent malarias are commonly used as *in vivo* experimental systems, since human malaria patients are less experimentally accessible. Here, the *P. yoelii* model could reveal the biological pathways that histone lactylation influences *in vivo*.

RNAseq analysis was previously conducted in groups of mice with early-stage versus late-stage *P. yoelii* infections (i.e., low versus high blood lactate) [14], and we re-analysed those data for the parasite (rather than the host) transcriptome, seeking changes that correlated with histone lactylation in late-stage infections. For comparison, we analysed the same data from *P. berghei* ANKA, where late-stage infections did not consistently generate hyperlactataemia [14].

All datasets were first corrected for different mixes of parasite developmental stages [20], and indeed marked differences in the proportions of schizonts were detected in early versus late-stage *P. yoelii* infections, while gametocytes were found in low proportions in early infections and were undetectable in late infections (S6 Table).

Differential expression (DE) analysis [21] performed on the corrected data revealed distinct expression profiles in late versus early stages of infection in both species (S12A Fig). Large cohorts of genes were differentially expressed (*P*-value ≤0.05): 502 genes in *P. yoelii* (238 upregulated and 264 downregulated) and 1521 genes in *P. berghei* (803 upregulated and 718 downregulated) (S12B Fig and S6 Table). (Using more stringent criteria, *P*-value ≤0.05 and |log2FC| ≥ 1, expression changes were similarly distributed across fewer total genes (S12B Fig). Late-stage *P. yoelii* parasites showed a strong downregulation of the lactate dehydrogenase (LDH) gene, as well as the glycolytic enzyme phosphofructokinase (PFK) and several genes putatively involved in gametocytogenesis. Several tRNA synthetases were also downregulated, with the interesting exception of isoleucine tRNA synthetase (isoleucine is the only amino acid that the parasite must scavenge from its host). Indeed, genes with the strongest expression changes were generally downregulated, while fewer – including several histones – were upregulated (Fig 8A and 8B). The opposite was true in late-stage *P. berghei*, where gene upregulation was most apparent, including the upregulation of phosphofructokinase and glyceraldehyde 3-phosphate dehydrogenase, and of factors involved in DNA replication and repair such as MCMs and RecQ helicase (Fig 8A and 8B).

We then sought to test whether a selection of genes whose expression changed in hyperlactataemic hosts were differentially marked with lactyl lysine. For this, we turned from the reanalysis of published transcriptomic data to the gene-specific analysis of newly-generated chromatin, harvested from *P. yoelii* parasites in mouse hosts with low or high blood lactate. We used chromatin immunoprecipitation (ChIP) to check whether a selection of *P. yoelii* genes that were differentially expressed in hyperlactataemic hosts were also differentially marked with lactyl lysine (Figs 8C and S12C). Indeed, lactylation in the coding sequences of all five chosen genes correlated strongly with blood lactate. These genes encoded histone H4 (PY17X_0944400), Ap2 transcription factor G (AP2G, PY17X_1440000) and three factors involved in metabolism (Lactate dehydrogenase, PY17X_1345100; ATP-dependent 6-phosphofructokinase, PY17X_0922000; and

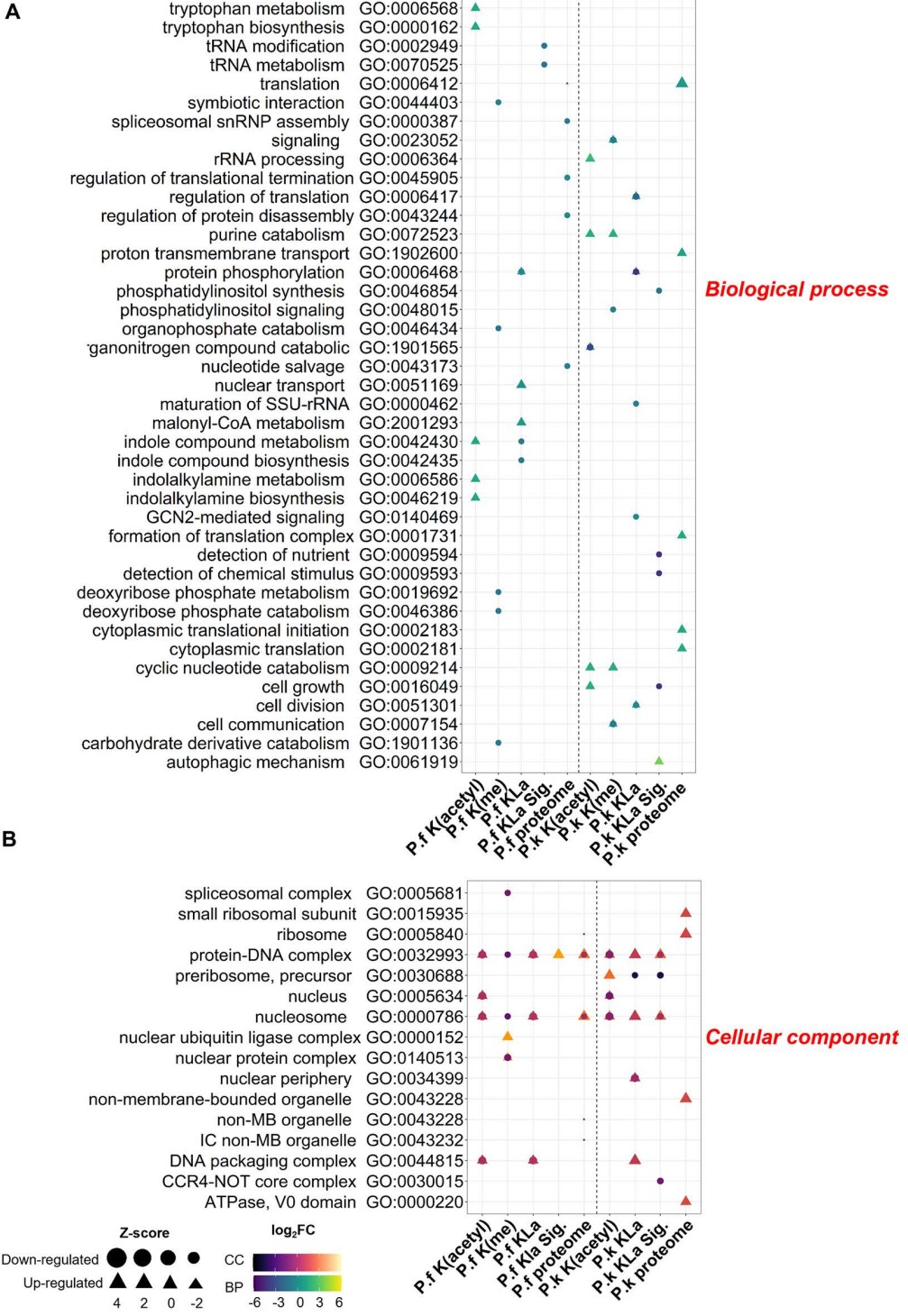

**Fig 7. Gene ontology (GO) enrichment analysis of biological processes and cellular components enriched in PTMs after lactate exposure.**
Charts represent the top 5 GO terms (biological processes **(A)**, cellular components **(B)**) with the highest P-value for each PTM. Triangles mark the GO terms associated with proteins whose PTMs were upregulated; circles mark those down-regulated. Shapes are color-coded based on the magnitude of the fold change. The size of the shape, represented by Z-score, indicates the magnitude of the likelihood that the term is up- or down-regulated. *P.f* and *P.k* represent *P. falciparum* (all data on the left of the vertical dashed line) and *P. knowlesi* (right of dashed line). For simplicity, K(me3) and K(me2) are both presented as K(me). 'KLa' refers to all KLa sites detected; KLa.Sig refers to lactylated proteins that changed significantly upon lactate exposure in trophozoites.

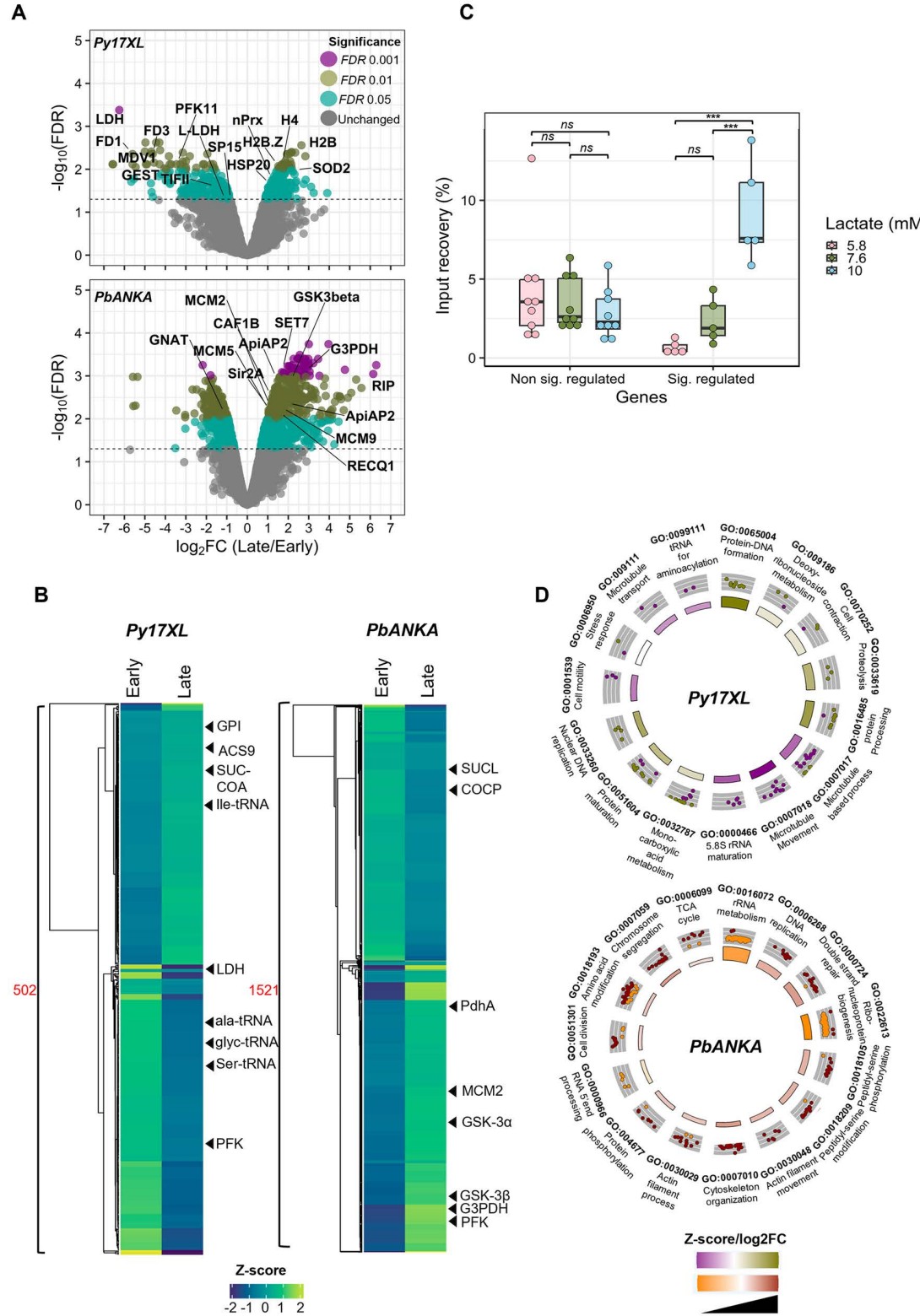

**Fig 8. In rodent malaria parasites *in vivo,* transcriptomic profiles and lactylation of representative genes both vary with levels of host lacta-taemia.** A: Volcano plots representing differentially expressed genes detected in late versus early infections of *P. yoelii* 17XL (top) and *P. berghei* ANKA (bottom). Data reanalysed from [14]. X-axes represent log$_2$ fold changes (FC) calculated from normalized CPM values; y-axes represent *P*-values.

Coloured dots represent DE genes, based on their significance threshold indicated by adjusted *P*-value (FDR) (grey dots represent those below any significance threshold). Horizontal and vertical dashed lines represent *P*-value and log$_2$FC significance thresholds. Gene name abbrevations in *P. yoelli*: SP15, transcription elongation factor SPT5; TIFII, eukaryotic translation initiation factor 3 subunit L; HSP20, HSP20-like chaperone; nPrx, peroxiredoxin; PFK11, ATP-dependent 6-phosphofructokinase; H2B, histone H2B; H4, histone H4; SOD2, superoxide dismutase [Fe]; GEST, gamete egress and sporozoite traversal protein; L-LDH, L-lactate dehydrogenase; LDH, lactate dehydrogenase; FD3, female development protein FD3; H2B.Z, Histone H2B variant; MDV1, male development protein 1; FD1, female development protein 1. Gene name abbrevations in *P. berghei*: ApiAP2, AP2 domain transcription factor; GSK3beta, glycogen synthase kinase-3 beta; GNAT, N-acetyltransferase in GNAT family; MCM2/5/9, DNA replication licensing factor MCM2/5/9; RECQ1, ATP-dependent DNA helicase RecQ1; SET7, histone-lysine N-methyltransferase SET7; G3PDH, glycerol-3-phosphate dehydrogenase; RIP, PIR protein; Sir2A, histone deacetylase Sir2A; CAF1B, chromatin assembly factor 1 subunit B. B: Heatmaps representing late and early transcriptional profiles in log$_{2CPM}$ (Z-score transformed) for DE genes (502 in *P. yoelii*, 1521 in *P. berghei*). Hierarchical clustering was performed to cluster genes' expression profiles with euclidean distance and ward.D clustering algorithm. C: Boxplot showing KLa enrichment detected via ChIP in the CDS of five *P. yoelii* genes whose expression significantly changed in response to hyperlactataemia (sig. regulated), and nine genes whose expression did not (non sig. regulated). Y-axis represents the percentage of chromatin input recovery (% IPed) for each ChIPed region. One biological replicate for each of three mice (i.e., three lactate conditions). Dots represent individual genes, horizontal line represents median, box represents interquartile range, whiskers represent variability outside the interquartile range (quartile -/+ 1.5 * IQR). Multiple comparison of means was performed using ANOVA and Tukey HSD post hoc tests at *p-value* < 0.05 for between-group comparison. Left boxplot (ANOVA, F = 0.788; d.fsum = 2; p-value = 0.466), right boxplot (ANOVA, F = 23.73; d.fsum = 2; p-value = 6.76e-05). *** p-value ≤ 0.001; ** p-value ≤ 0.01; *p-value ≤ 0.05; ns, not significant. D: Biological process gene ontology (GO) enrichment analysis of DE genes in *P. berghei* and *P. yoelii*. Plots show the top 15 GO terms with the highest *P*-values. The outer circle displays scatterplots of the expression levels (log$_2$FC) for the genes (dots) in each GO term, whereas the inner circle is a bar plot where the height of the bar indicates the significance of the term (−log10 *P*-value). The colour corresponds to the Z-score, which indicates whether the term is likely to be decreased (negative value) or increased (positive value).

isoleucine-tRNA ligase (Ile-tRNA), PY17X_1442500). By contrast, there was no correlation between blood lactate and lactylation of chromatin in nine control genes that did not show differential expression. Lactylation within 1kb upstream of the same five genes was more variable and less strongly correlated with blood lactate than lactylation within the gene body (S12D Fig). Thus, it is likely that lactylation of histones and/or other chromatin proteins within gene bodies directly regulates gene transcription, at least in the *P. yoelii* model.

When GO enrichment analysis was conducted on the up- and down-regulated genes, it was clear that late-stage *P. yoelii* parasites activated biological (Fig 8D) and molecular (S13 Fig) functions including proteolysis and mono-carboxylic acid metabolism, while late-stage *P. berghei* parasites activated contrasting functions including DNA replication and repair, chromosome segregation, cell division and actin/cytoskeleton dynamics. Amongst functional pathways, late-stage *P. yoelii* parasites upregulated fatty acid metabolism whereas *P. berghei* upregulated sugar metabolism and glycolysis (S13 Fig). Thus, it appeared that *P. yoelii* parasites in hyperlactaemic hosts showed features of 'dormancy'. This phenomenon can be induced in cultured *P. falciparum* by isoleucine starvation and is characterised by slowing of the replicative cycle and proteolysis [22]. By contrast, *P. berghei* – in hosts without hyperlactataemia – conversely upregulated pathways involved in replication.

## Discussion

This work represents the first characterisation of a new epigenetic mark, histone lactylation, in the malaria parasite *Plasmodium.* A lactylation epigenetic pathway could be highly influential in the biology and pathology of malaria, because some *Plasmodium* species, including *P. falciparum*, characteristically cause hyperlactataemia in their hosts [3]. It would therefore make sense if these parasites had evolved to sense and respond to hyperlactataemia, potentially modulating their virulence or transmission dynamics. Furthermore, 'epigenetic drugs' such as histone methyltransferase or deacetylase inhibitors have been shown to kill parasites, so epigenetic pathways in general are potential targets for antimalarial drug development [23,24].

Here, we show that histone lysine lactylation in several *Plasmodium* species is inducible by exogenous lactate. We detected histone lactylation across four *Plasmodium* species, both *in vitro* and *in vivo*, but it was not equally prominent, nor equally responsive to exogenous lactate, in all species. Rather, it was more abundant and more inducible in species that cause hyperlactataemia: *P. falciparum* and *P. yoelii*. In cultured *P. knowlesi*, a species only rarely associated with

lactic acidosis in humans [25], the modification was less abundant and was not clearly regulated across a normal cell cycle, as it was in *P. falciparum.* In *P. berghei* ANKA, which did not cause consistently high levels of hyperlactataemia in mice, the modification was not clearly responsive to blood lactate, whereas in *P. yoelii* 17XL, it was. Commensurate with profound species-specific differences, mass spectrometry showed that the histone lysine residues lactylated in *P. falciparum* and *P. knowlesi* were generally different.

In human cells, it has been suggested that lactylation may act like acetylation on histone tails – as a transcription-activating mark [10]. In *Plasmodium*, this may also be true. KLa was detected throughout nuclear chromatin, which is mainly euchromatic, with no specificity for heterochromatin. Furthermore, Kme marks are generally gene-silencing in *P. falciparum*, and mass spectrometry showed that these were mostly downregulated after lactate exposure, suggesting reciprocal regulation of KLa-induced activation and Kme-induced silencing. Again, species-specificity was apparent: such reciprocity was notably absent in *P. knowlesi.*

KLa and Kac were not, however, interchangeable: only a small subset of Kac sites in either species was detected as lactylated, and a large proportion of the KLa sites, but only a few Kac sites, were induced by exogenous lactate. This has implications for the theory that histone acetyltransferases (HATs), which normally use acetyl-CoA, can also use lactyl-CoA to act as writers of KLa [10]. Since lysine residues were not simply modified with either moiety according to substrate abundance, our results would suggest that only some HATs might use lactyl-CoA, on only some lysine sites. The possible contributions of *P. falciparum* HATs to lactyltransferase activity are currently being investigated. An entirely separate pathway was also recently reported, in which alanine tRNA synthetase moonlights as a lactyltransferase [26,27] and this might also operate in *Plasmodium.*

Besides histone tail lactylation, we detected numerous non-histone proteins being lactylated. These were certainly not comprehensively surveyed because our mass spectrometry focussed on small, chromatin-associated proteins (large, nucleoplasmic or cytoplasmic proteins were probably under-detected). Nevertheless, we detected lactylation on ApiAP2 transcription factors, RNA processing factors, HATs and HDACs amongst others. Unlike the histone tails, where almost every lactylated lysine could also be acetylated, some non-histone proteins were uniquely lactylated (and again, they were largely species-specific between *P. falciparum* and *P. knowlesi*). Thus, the lactyl PTM may have evolved, species-specifically, to regulate *Plasmodium* biology on at least three levels: the epigenetic control of gene transcription via histone tail marks, the co-transcriptional control of RNA processing, and the post-translational modulation of protein activity in HATs, HDACs, etc. – potentially biasing their enzyme activity from acetyl- to lactyl- specificity. The biological pathways that are actually regulated await investigation, because lactylated proteins were associated with a very wide range of GO terms. Important pathways could include stress-responsive transcriptional programmes controlled by ApiAP2s (possibly including gametocyte conversion, since lactylated ApiAP2G was detected in *P. falciparum*, albeit at a low level), and the expression of *var* genes or other virulence gene families that are epigenetically regulated via HAT/HDAC activity [8].

To gain insight into biological changes that can occur in parasites exposed to hyperlactatemia *in vivo*, we analysed transcriptomic data from *P. yoelii* in hyperlactataemic mice. Profound changes were detected, with features of metabolic dormancy – a programme previously reported in *P. falciparum* parasites starved of isoleucine, the only amino acid that malaria parasites cannot gain from haemoglobin catabolism [22]. Starved *P. falciparum* parasites slowed their developmental cycle and activated proteolysis: features also seen here in the GO analysis for late-stage *P. yoelii*, but not in late-stage *P. berghei* where the hosts are not generally hyperlactataemic and the parasites do not induce high levels of histone lactylation. Five genes that were differentially expressed in *P. yoelii* during hyperlactataemia were differentially marked with lactyl lysine, strongly suggesting a direct transcription-modulating effect of lactylation – on histones and/or other chromatin proteins. Future work will focus on comprehensive analyses of the transcriptome and epigenome, both in human parasites in culture and in rodent parasites *in vivo*, to establish which transcriptional programmes in parasites exposed to hyperlactataemia are actually epigenetically controlled by histone lactylation, or by lactylation of other chromatin proteins.

## Methods

### Ethics statement

All experimental protocols and procedures were performed in accordance of the UK Animals (Scientific Procedures) Act (PP8697814), and were approved by the Imperial College Animal Welfare and Ethical Review Board (for *P. yoelii*) or the University of Cambridge AWERB (for *P. berghei*). The Office of Laboratory Animal Welfare Assurance for the University of Cambridge covers all Public-Health-Service-supported activities involving live vertebrates (no. A5634-01). This study was carried out in compliance with the ARRIVE guidelines (https://arriveguidelines.org/). All studies were in accordance with the Laboratory Animal Science Association's guidelines for good practice. Prior to any experimental interventions, all mice were acclimatized to animal facilities for one week.

### Culture and synchronisation of *P. falciparum* and *P. knowlesi*

*Plasmodium falciparum* (NF54 strain) and *Plasmodium knowlesi* (A1-H.1 strain) were grown in RPMI 1640 media (Sigma, R4130) with 2.3 g/L sodium bicarbonate, 50 mg/L hypoxanthine (Sigma, H9377) and 25 µg/L gentamicin (Melford Laboratories, G38000-1), supplemented with 5 g/L Albumax II (Invitrogen, 11021037), 10% horse serum (Gibco, 10368902) and 4 g/L glucose for *P. knowlesi*, or with 2.5 g/L Albumax II and 5% human serum for *P. falciparum*. *P. falciparum* and *P. knowlesi* were cultured in human red blood cells (NHS Blood and Transplant) at 4% or 2% haematocrit respectively in 3% oxygen, 5% $CO_2$ and 92% nitrogen gas mixture at 37°C.

For *P. falciparum* synchronization, parasites at ~ 6% parasitaemia were initially synchronised with 5% sorbitol (w/v in water) and allowed to grow to late schizont stage. ML-10 (LifeArc) [28,29] was then added for 2h at a final concentration of 75 nM. Schizonts were purified via a 65% Percoll (GE Healthcare) gradient (v/v in PBS), washed twice with complete media, and allowed to reinvade in 25% haematocrit in 5 ml of complete media in the abovementioned gas mixture at 37°C for 3h with agitation at 220 rpm. 5% sorbitol was then used to remove residual schizonts and produce a tightly synchronised culture (0–3 hours post invasion (hpi)).

For *P. knowlesi* synchronization, parasites at ~ 6% parasitaemia were centrifuged through a 55% Nycodenz gradient (v/v in RPMI) from 100% Nycodenz stock pH 7.5 (27.6% w/v Nycodenz, 5 mM Tris HCl, 3 mM KCl, 0.3 mM $CaNa_2 \cdot EDTA$). Ring stages isolated in the bottom layer were allowed to grow to late schizont stage. ML-10 was then added to a final concentration of 150 nM for 2h. Parasites were centrifuged through a second Nycodenz gradient to collect schizonts from the top layer, which were washed twice with complete media and allowed to reinvade as above. A second Nycodenz cleanup was then performed to remove residual schizonts and produce a tightly synchronised culture (0–3 hpi).

### Growth of *P. yoelii* and *P. berghei*

Eight-week-old wild-type female C57BL/6J or CD1 mice were obtained from Charles River Laboratories. All mice were specified pathogen-free, housed in groups of five in individually ventilated cages, and provided ad libitum access to food and water.

*P. yoelii* (17XL strain) was serially passaged through C57BL/6J mice, following previously described protocols [14]. Blood was collected via aseptic cardiac puncture from infected donor mice under non-recovery isoflurane anesthesia and diluted in sterile phosphate-buffered saline to achieve the desired parasite concentration. Experimental mice were subsequently infected with $10^6$ live parasites via intraperitoneal (i.p.) injection.

*P. berghei* (ANKA 2.34 strain) was maintained by serial passage through CD1 mice as in previously described protocols [30]. Experimental mice were pre-treated with 150 µl of 6 mg/ml phenylhydrazine i.p. three days prior to infection, to induce reticulocytosis. Blood from donor mice at 1–5% parasitaemia was collected by cardiac puncture under non-recovery Ketaset/Rompun anaesthesia and used to infected experimental mice by passage of 150 µl blood, i.p..

Tail capillary blood samples were collected to prepare blood smears for parasitemia assessment and lactate measurement. Parasitemia was quantified through microscopy of thin blood smears stained with 10% Giemsa, following established protocols [14]. Heparinized blood was collected via cardiac puncture under non-recovery anesthesia, into syringes pre-loaded with heparin, and red blood cells were separated from plasma by centrifugation at 4,000g for 10 minutes. Levels of lactate in the cardiac blood of each mouse were immediately measured using the Lactate Pro 2 meter (HAB Direct) for *P. yoelii* experiments, or the L-lactic acid (L-Lactate) assay kit (Megazyme K-LATE) for *P. berghei* experiments. Red blood cells were harvested for parasite histone extraction from ten mice per rodent malaria species, of which seven yielded a viable amount of parasite material when infected with *P. yoelii* 17XL and five yielded an appropriate amount when infected with *P. berghei* ANKA 2.34.

## Sodium L-lactate treatment of *P. falciparum* and *P. knowlesi* cultures

For titration experiments, *P. knowlesi* at 12–14 hpi and *P. falciparum* at 24–26 hpi, both at 1% parasitaemia, were washed into fresh media and then treated with 0mM, 5mM, 10mM or 25mM sodium L-lactate (Sigma), for 8h (*P. knowlesi*) and 12h (*P. falciparum*). Subsequently, samples were harvested and crude histone extractions were performed (as described below).

For mass spectrometry on histones from mixed-stage cultures of both species, parasites at a parasitaemia of ~ 6% were washed into fresh media to remove any endogenously-produced lactate and sodium L-lactate (Sigma), dissolved in incomplete media, was added to a final concentration of 25mM. For mass spectrometry on histones from trophozoites, *P. falciparum* at 24–26 hpi and *P. knowlesi* at 12–14 hpi were treated with sodium L-lactate. Treatments were for 12 and 9h on the two species respectively.

## Extraction of *Plasmodium* histones

*P. falciparum* and *P. knowlesi* cultures were collected by centrifugation at 800 *x g* for 5 minutes, followed by resuspending the pellet in 1.5 volumes 1X PBS. To release parasites from erythrocytes, saponin was added to a final concentration of 0.1% and incubated on ice for 10 minutes, followed by centrifugation at 12,000 *x g* at 4°C for 10 minutes. Parasite pellets were washed three times by adding ice cold 1X PBS (2 volumes of the saponin lysate) followed by centrifugation at 6000 *x g* for 2 minutes.

For *P. yoelii* and *P. berghei,* infected blood obtained by mouse cardiac puncture was placed in 50 mL RPMI HEPES modification media ('schizont media' - no FCS) and spun at 800 *xg* for 5 minutes, followed by an RPMI wash and centrifugation at the same speed. To remove host leukocytes, blood was passed through Plasmodipur filters (EuroProxima, 8011) using a syringe, followed by washing the filter with RPMI media to elute any remaining blood. The filtration procedure was repeated a second time before collecting the blood pellet by centrifugation at 800 *xg* for 5 minutes. Filtered blood was then mixed with 0.1% saponin on ice for 15 minutes to release parasites, which were then collected as above.

Parasites from all species were processed either for crude or acid-based histone extraction. For crude histone extraction, parasite cell pellets were lysed by adding 1X RIPA buffer (250mM Tris-Hcl pH 8.0, 750 mM NaCl, 5% NP-40, 5% sodium deoxycholate, 0.5% SDS, 1mM phenylmethylsulfonyl fluoride, 1X cComplete EDTA-free protease inhibitor cocktail (Roche)) followed by three freeze/thaw cycles on dry ice, 2 minutes each. Laemmli buffer was then added to a final concentration of 1X, boiled at 95°C for 10 minutes and stored at -80°C for western blot analysis.

For mass spectrometry analysis, nuclei were liberated from parasite cell pellets by gently resuspending the parasite pellet twice in two volumes of hypotonic buffer A (10 mM Tris-HCl (pH 8.0), 3 mM MgCl$_2$, 0.2% v/v Nonidet P-40, 0.25 M sucrose, and a cocktail of EDTA-free protease inhibitors (Roche, 0469313200) followed by centrifugation at 4000 *x g* at 4°C for 10 minutes. The resulting chromatin pellet was then resuspended in two volumes of hypotonic buffer B (10 mM Tris-HCl, pH 8.0, 0.8 M NaCl, 1 mM EDTA [including protease inhibitor cocktail]) and incubated on ice for 10 minutes

followed by centrifugation at 4000 x $g$ at 4°C for 10 minutes. The chromatin pellet was then resuspended with eight volumes of 0.25 M ice-cold HCl followed by vigorous vortexing and rotation for 2h at 4°C. Acid-soluble recovered proteins in the supernatant were collected by centrifugation at 12,000 $x$ $g$ for 30 minutes at 4°C. Histone-containing supernatant was mixed with an equal volume of 30% TCA and incubated overnight rotating at 4°C, followed by centrifugation at 12,000 $x$ $g$ for 15 minutes at 4°C to collect the histone-containing pellet. This was then washed with 500 µl of ice-cold acetone and incubated at -20°C for 3h, followed by centrifugation at 12,000 $x$ $g$ for 15 minutes. Histone pellet was then air-dried and reconstituted in 50 µl ddH$_2$0. Protein concentration was measured using Qubit protein quantification kit (Invitrogen, Q33211). Purity of extracted histones was then assessed by SDS-PAGE on a 15% acrylamide gel stained with Coomassie blue G-250.

### Western blotting

Protein extracts were loaded on 15% SDS polyacrylamide gels. For lactate-titration blots, approximately 10 µg and 100 µg were loaded for *P. falciparum* and *P. knowlesi* protein extracts respectively. For validation of histone lactylation prior to mass spectrometry, ~10 µg of acid-extracted histones were used. All gels were run at 120 V for 3h and transferred to 0.45 µM nitrocellulose membrane (GE healthcare, 10600020). For both chemiluminescent (Figs 1–3) and fluorescent (S4 Fig) blots, membranes were blocked in PBS-0.1% Tween 20 containing 5% bovine serum albumin, then probed with designated primary antibodies. For chemiluminescent detection, horseradish peroxidase (HRP)-coupled secondary antibodies were used and developed with ECL Plus Western Chemiluminescent HRP Substrate (ThermoScientific, 11527271). For fluorescent detection, fluorescent secondary antibodies were used. Blots were imaged with an Azure biosystems imager 500 and densitometry was performed with ImageJ for all western blots from titration and timecourse experiments. All lactylated histone signals were normalised using a parallel blot for total histone H4 (except for the fluorescence blot in S3 Fig, where lactylated histones were normalized using the H4 signal on the same blot).

### Lactate assay

Lactate assays were performed using the L-lactic acid (L-Lactate) assay kit (Megazyme K-LATE). Media samples were taken from 5 ml cultures of *P. falciparum* NF54 and *P. knowlesi* A1-H.1, synchronised with Percoll at the ring stage, starting with 0.5% parasitaemia, every 24 h for 5 days. For two biological repeats the cultures were split every 48 h and for another two biological repeats the cultures were not split. The lactate assay was performed according to the manufacturer's instructions (Megazyme K-LATE) on triplicate media samples of 2µl, plated in the 96-well format. The assays were read on a FLUOstar Microplate Reader (BMG LABTECH). Parasitaemia was verified by microscopy every time samples were collected. Results were analysed using GraphPad Prism.

### Immunofluorescence assay

Smears were made from *P. falciparum* 3D7 HP1_HA + BSD (gift from Prof Till Voss [16]) and NF54, using mixed stage cultures at 4% parasitaemia. Smears were air dried, fixed in 4% paraformaldehyde/PBS for 10 minutes, washed in PBS, air dried and stored at 4°C. Parasites were permeabilised in 0.2% Triton X-100 for 15 minutes, washed in PBS, and blocked in 1% BSA/PBS with 0.1% Tween-20 for 1 h. Primary antibody labelling was done for 1 h, diluted 1:500 in 1% BSA/PBS, followed by three 5-minute washes with block (1% BSA with 0.1% Tween-20). Secondary antibody labelling was done for 1 h, diluted 1:1000 in 1% BSA/PBS, followed by one 5-minute wash with block (see Table 1 for antibodies). The slides were then incubated with DAPI (4'6-diamidino-2-phenylindole) (Thermo Fisher Scientific) at 2 µg/mL in PBS for 5 minutes, followed by one 5-minute wash with block. All incubation and labelling steps were done at room temperature. Slides were cured using ProLong Diamond Antifade Mountant (Invitrogen) at room temperature and stored at 4°C prior to visualisation. Slides were imaged using a confocal fluorescence microscope Zeiss LSM700. Images were pseudo-coloured and merged using Fiji-win64 (Image J).

**Table 1. Antibodies used in western blotting and immunofluorescence microscopy.**

| Primary antibody | Dilutions | | Secondary antibody | Dilutions | |
| --- | --- | --- | --- | --- | --- |
| | WB | IF | | WB | IF |
| HA - Anti-HA High Affinity Rat mAB 50 ug (Roche 11867423001) | | 1:500 | Alexa Fluor 488-conjugated anti-rat IgG (ThermoFisher A-11006) | | 1:1000 |
| KLA - Anti-L-Lactyllysine Rabbit mAb 100 ul (PTM-1401) | 1:1000 (chemi) 1:10,000 (fluor) | 1:500 | Alexa Fluor 594-conjugated anti-rabbit IgG (ThermoFisher A-11012) Goat-anti-rabbit IgG IR800 (AzureSpectra, AC2134) | 1:10,000 | 1:1000 |
| H4KL12 - Anti-Lactyl-Histone H4 (Lys12) Rabbit pAB 100 ul (HOLZEL PTM-1411) | 1:000 | 1:500 | Alexa Fluor 594-conjugated anti-rabbit IgG (ThermoFisher A-11012) | | 1:1000 |
| H3ac - Ant-Histone H3ac (pan-acetyl) Rabbit pAb 100 μg (Active motif 61637) | 1:4000 | 1:500 | Alexa Fluor 594-conjugated anti-rabbit IgG (ThermoFisher A-11012) | | 1:1000 |
| H4ac - Anti-Histone H4 (acetyl K5+K8+K12) 50 μg (Abcam ab233193) | 1:4000 | 1:500 | Alexa Fluor 594-conjugated anti-rabbit IgG (ThermoFisher A-11012) | | 1:1000 |
| Rabbit anti-histone H4 (Abcam, ab10158) | 1:3000 | 1:500 | HRP-conjugated goat anti-rabbit IgG (Abcam, ab6721) | 1:2500 | 1:1000 |
| Mouse anti-histone H4 (Abcam, ab17036) | 1:1000 (fluor) | | Goat-anti-mouse IgG IR700 (AzureSpectra, AC2129) | 1:10,000 | |

## Mass spectrometry

**Sample preparation for mass spectrometry.** 50 μg of acid-extracted histones were resolved on SDS-PAGE 15% acrylamide gel (95 V, 3 h). Gel was stained with QC colloidal Coomassie blue G-250 (Biorad, 1610803) for 3 h, then washed with several changes of nuclease-free water over 1 h. Bands between 10 and 15 kDa were then excised and washed with several changes of 50% acetonitrile for destaining, followed by 100% acetonitrile. Gel plugs were then incubated in 0.05 M $NH_4HCO_3$ and 10 mM DTT at 56°C for 45 minutes for disulphide bond reduction, and with 55 mM iodoacetamide and 0.1 M $NH_4$ for alkylation, followed by treatment with iodoacetamide for 30 min at 30°C in the dark. Gel plugs were then washed with water and 50% acetonitrile. For dehydration, 100% acetonitrile was added, followed by rehydration in 50 mM $NH_4HCO_3$ containing 12.5 ng/μL trypsin (Promega modified trypsin). Samples were incubated overnight at 37°C. After digestion, the supernatant was pipetted into a sample vial and loaded onto an autosampler for automated LC-MS/MS analysis.

**Reverse phase chromatography and mass spectrometry.** All LC-MS/MS experiments were performed using a Dionex Ultimate 3000 RSLC nanoUPLC system (Thermo Fisher Scientific Inc, Waltham, MA, USA) and a Q Exactive Orbitrap mass spectrometer (Thermo Fisher Scientific Inc, Waltham, MA, USA). Separation of peptides was performed by reverse-phase chromatography at a flow rate of 300 nL/min and a Thermo Scientific reverse-phase nano Easy-spray column (Thermo Scientific PepMap C18, 2mm particle size, 100A pore size, 75 mm i.d. x 50 cm length). Peptides were loaded onto a pre-column (Thermo Scientific PepMap 100 C18, 5mm particle size, 100A pore size, 300 mm i.d. x 5mm length) from the Ultimate 3000 autosampler with 0.1% formic acid for 3 minutes at a flow rate of 15 mL/min. After this period, the column valve was switched to allow elution of peptides from the pre-column onto the analytical column. Solvent A was water + 0.1% formic acid and solvent B was 80% acetonitrile, 20% water + 0.1% formic acid. The linear gradient employed was 2–40% B in 90 minutes. Further wash and equilibration steps gave a total run time of 120 minutes.

The LC eluant was sprayed into the mass spectrometer by means of an Easy-Spray source (Thermo Fisher Scientific Inc.). All m/z values of eluting ions were measured in an Orbitrap mass analyzer, set at a resolution of 35000 and was scanned between m/z 380–1500. Data dependent scans (Top 20) were employed to automatically isolate and generate fragment ions by higher energy collisional dissociation (HCD, NCE:26%) in the HCD collision cell and measurement of the resulting fragment ions was performed in the orbitrap analyser, set at a resolution of 17500. Singly charged ions and ions

with unassigned charge states were excluded from being selected for MS/MS and a dynamic exclusion window of 40 was employed.

**Data processing, protein quantification and PTM localisation.** For all experiments, raw mass spectrometry data were analyzed using MaxQuant [31] v. 1.5.0.0 by employing Andromeda for MS/MS spectra search against the *P. falciparum* and *P. knowlesi* PlasmoDB FASTA proteome reference (release 64), and a common contaminant database [32]. In MaxQuant, enzyme specificity was set to Trypsin/P; carbamylation of cysteines was set as a fixed modification and Acetyl lysine, lactyl lysine, and di/tri methyl lysine, as variable modifications. Missed cleavage sites were set to 5 with a maximum number of 3 modifications per peptide. For protein quantification, major protein aggregation was changed to sum. Match between run was turned on. PTMs quantification files generated by MaxQuant were exported to Perseus (v3.6.2) [33] to collapse localisation site information and remove reverse and contaminants. For all PTMs, sites were identified in both conditions for mixed-stage experiments, or in at least two replicates per condition for trophozoites. Remaining protein quantification and PTM quantification steps were performed in R (v3.6.2) as described below.

**Downstream data analysis for differential protein and PTM expression.** For all experiments, PTM peptide quantification files obtained after Perseus filtering steps described above, and ProteinGroup.txt protein quantification files obtained by MaxQuant, were exported to R for further processing. For mixed-stage experiments, $log_2$ transformed PTM intensities were normalised using quantile normalisation using the normalizeBetweenArrays function from limma package (v3.42) [34]. For trophozoite-stage experiments, $log_2$ transformed PTM intensities were normalised using quantile and VSN normalisation for *P. falciparum* and *P. knowlesi*, respectively, using the normalizeBetweenArrays function from limma package (v3.42). VSN was chosen for the latter due to technical variability between biological replicates, which affected replicate 1 only from both + and - lactate conditions. Imputation of missing values was done by low-rank approximation of the normalised PTM data matrix using msImpute [35]. In brief, imputation was performed with V2 method if values were missing at random, or by Barycenter approach if missingness was not at random. Differentially expressed sites were calculated using limma (two-sided, BH $p$ value < 5%). At the protein level, reverse and contaminants were removed from ProteinGroup.txt quantification file, followed by extracting LFQ intensities and requiring at least two values per condition. For better accuracy in global protein levels, spectraCounteBayes function from DEqMS package (v1.22.0) [36] was used to make limma's empirical-Bayesian variance calculation dependent on the number of detected peptides/PTMs per protein generated by MaxQuant, rather than estimating a fixed prior distribution for all proteins. Finally, differentially expressed proteins were calculated using limma (two-sided, BH FDR < 1%).

## RNA-seq data analysis

Raw reads from the RNA-Seq experiment generated by [14], belonging to Py17XL and PbANKA early and late infection stages, were downloaded, trimmed and quality checked by Trimmomatic v.0.35 [37]. Low quality reads with Phred quality score values lower than 30 and lengths less than 20 bp were removed. (https://www.bioinformatics.babraham.ac.uk/projects/fastqc/). Filtered reads were further quality controlled by FastQC v.0.11.7 (https://www.bioinformatics.babraham.ac.uk/projects/fastqc/) and used for transcript quantification using SALMON v.0.82 [38]. Py17XL and PbANKA reference transcript datasets were indexed using the quasi-mapping mode (--type quasi) of Salmon to create k-mers of lengths 31 (-K 31). For transcript quantification, input data sequence bias correction and bootstrapped abundance estimate were performed using the "seqBias" and "nom-Bootstraps 30" parameters from Salmon, respectively. Read counts and transcript per million reads (TPMs) were generated using tximport R package version 1.10.0 and lengthScaledTPM method [39] with inputs of transcript quantifications from tool SALMON. Low-expressed transcripts and genes were filtered based on analysing the data mean-variance trend. The expected decreasing trend between data mean and variance was observed when expressed transcripts were determined as those that had 3 of the 6 samples with count per million reads (CPM) 1, which provided an optimal filter for low expression. A gene was considered to be expressed if any of its transcripts with the above criteria was expressed. The TMM method was used to normalise the gene and transcript read counts to –CPM [40].

Prior to DE analysis and to control for heterogeneity within parasite developmental stages, we performed parasite stage deconvolution as follows. A signature matrix was taken from [20], which contains CPM-normalised gene expression measurements calculated across different parasite stages (male and female gametocyte, trophozoite, ring, merozoite, and schizont stages). *P. berghei* and *P. yoelii* gene names were converted to one-to-one orthologous *P. falciparum* gene names using an orthologous gene file obtained from the Malaria Cell Atlas. CPM-normalised counts alongside the signature matrix were then uploaded to CIBERSORT for deconvolution analysis using 100 permutations and disabling quantile normalisation (as recommended by CIBERSORT) to obtain the parasites cell proportions. This was performed separately for reads mapping to *P. berghei* and *P. yoelii*.

For DE analysis, limma suite was used [34,41] by fitting a linear model accounting for the cell proportions obtained in CIBERSORT. To compare the expression changes between conditions of experimental design, the contrast groups were set as Late-Early. For DE genes/transcript, the fold change of gene/transcript abundance were calculated based on contrast groups and significance of expression changes were determined using t-test. P-values of multiple testing were adjusted with BH to correct false discovery rate (FDR) [42]. A gene/transcript was considered significantly DE in a contrast group if it had adjusted p-value < 0.01 and |log2FC| ≥ 1.

## Gene ontology (GO) pathway enrichment analysis

For RNA-Seq datasets, functional GO and pathway enrichment analysis were performed on significantly DE genes only, using PlasmoDB database [43] for annotation with default parameters. The GO terms (biological process, molecular function and cellular component) and pathways were identified to provide biological insights into the significance of DE using a *P-value* ≤ 0.05. For visualization of GO and pathways terms, combined with their associated gene expression profiles/ significance, GOPlot R package v.1.02 was used [44].

For mass-spectrometry, functional GO enrichment analysis was performed using PlasmoDB database for annotation, with default parameters, on significantly differential proteins of the acid-extracted proteome and PTMs (for the trophozoite-stage experiment only). Exceptionally, for the KLa PTM, GO enrichment analysis was performed on both significantly and non-significantly changing lactylated proteins. Bubble plots were generated using GO terms combined with their associated genes expression profiles/significance across all PTMs and the acid-extracted proteome via in-house R scripts. For both datasets,

$$Z-\text{score} = \frac{(Up - Down)}{\sqrt{Count}}$$

Where *up* and *down* are the number of genes up-regulated (logFC > 0) in the data or down-regulated (logFC < 0), respectively.

## Chromatin immunoprecipitation (ChIP) and quantitative PCR (qPCR)

Blood was harvested from three mice infected with *P. yoelii 17XL* at different levels of parasitemia and lactatataemia. Purification of erythrocytes from host leukocytes and release of parasites were performed as described above. Parasites were crosslinked immediately with 1% paraformaldehyde (Sigma) by rotating for 10 mins at 37°C, then quenching with 0.125 M glycine for 5 mins at room temperature. Nuclei were liberated by gently resuspending the parasite pellet twice in two volumes of hypotonic buffer A (10 mM Tris-HCl (pH 8.0), 3 mM MgCl$_2$, 0.2% v/v Nonidet P-40, 0.25 M sucrose, EDTA-free protease inhibitors (Roche, 0469313200)) and holding on ice for 15 mins. This was followed by 15 strokes using a dounce homogenizer and centrifugation at 4000 x g at 4°C for 10 mins. The chromatin pellet was then resuspended in two volumes of hypotonic buffer B (10 mM Tris-HCl, pH 8.0, 0.8 M NaCl, 1 mM EDTA, protease inhibitor cocktail) and incubated on ice for 10 mins followed by centrifugation at 4000 x g at 4°C for 10 mins. 2–3 µg of chromatin were resuspended in 50

μl of shearing buffer (1% SDS, 50 mM Tris pH 8.0, 10 mM EDTA, 1X protease inhibitor cocktail) and sheared for 47 mins using an M220 sonicator (Covaris) at 5% duty cycle, 200 cycles per burst, and 75 W of peak incident power to generate 150–400 bp fragments.

Sheared chromatin was diluted 2x in ChIP dilution buffer (0.01% SDS, 1.1% Triton X-100, 1.2 mM EDTA pH 8.0, 25 mM Tris HCL pH 8.1, 150 mM NaCl, 1X protease inhibitor cocktail) and pre-cleared by incubating with protein A agarose beads (Thermoscientific, 20365) for 1 h at 4°C. 10% of the cleared chromatin was reserved as the input control and 90% was incubated for 4 h at 4°C using 0.75 μg of anti-KLa antibody coupled with protein A agarose beads. After one wash with low salt buffer (0.1%SDS, 1% Triton X-100, 2 mM EDTA, 20 mM Tris-HCL pH 8.1, 150 mM NaCl), one with high salt buffer (0.1% SDS, 1% Triton X-100, 2 mM EDTA pH 8.0, 20 mM Tris-HCl pH 8.0, 500 mM NaCl), and one with lithium chloride (0.25 M LiCl, 1% NP-40, 1% sodium deoxycholate, 1 mM EDTA, 10 mM Tris-HCl pH 8.1), the immunoprecipitate was eluted twice in 85 μl elution buffer (1% SDS, 0.1M NaHCO$_3$) by rotation at 65°C. ChIPed and input DNA were incubated with 2 μg RNase A (Roche, 10109169001) at 37°C for 30 min with gentle shaking. Samples were then resuspended in decrosslinking buffer (150 mM NaCl, 0.2% SDS, 600 μg Proteinase K (Ambion, AM2546)) by incubation at 65°C for 3 h with interval shaking. DNA was purified overnight at -20°C by adding 0.1 volume of 3M sodium acetate, 4 μl of linear polyacrylamide (Ambion, AM9520) and 2.5 volumes of ice cold ethanol, and collected by centrifugation at 13,000 xg for 30 mins at 4°C. DNA pellets were washed once with 70% ice cold ethanol and resuspended in 20 μl of nuclease free water.

qPCR was conducted with 3–4 ng of DNA in technical duplicate, using a QuantStudio 6 Pro Real-Time PCR System: 2 mins denaturation at 95°C, 40 cycles of 5s 95°C, 30s 60°C. Data were presented as the percentage of DNA immunoprecipitated relative to input ($100*2^{(Ct_{input}-Ct_{eluates})}$). Primers were designed in the CDS and within 1kb upstream of the TSS using Primer3 and are listed in Table 2.

## Supporting information

**S1 Fig. *Plasmodium berghei* histones are not inducibly lactylated in hyperlactaemic mice.** Lysine lactylation was measured in *P. berghei* ANKA parasites exposed to varying levels of lactate in the blood of the host mouse. Western blots show 'Pan KLa' and histone H4 as a control. M1-M5, individual mice; L, ladder. Scatter plot represents Pearson correlation ($R^2 = 0.06$, p > 0.05) between histone KLa signal intensities normalized to H4 (*y*-axis) and lactate levels in the blood (*x*-axis) upon parasite collection from host.
(PDF)

**S2 Fig. Histone lactylation is distributed throughout the nucleus.** A: Confocal microscopy images showing the nuclear locations of acetyl histone H4 (H4ac) and of HP1 in *P. falciparum* 3D7 HP1_HA. H4ac, red; HP1_HA, green; DAPI, blue. B: Confocal microscopy images showing the location of KLa in *P. falciparum* NF54. This confirms that the location is the same as that shown in main Fig 4, and is not affected by the tagging of HP1 in the line 3D7 HP1_HA (strain used in main Fig 4). KLa, red; DAPI, blue. C: Images as in (B), showing lactyl-H4K12 in the *P. falciparum* NF54 strain. D: Images as in (B), showing H4ac in the *P. falciparum* NF54 strain. Scale bar (2 μm) applies to all images. R, ring stage, T, trophozoite stage; S, schizont stage.
(PDF)

**S3 Fig. Acid based histone extraction: quality control.** A: Coomassie-stained gel of ~ 25 μg of acid-extracted histones with (+)/without (-) lactate (lac) resolved between ~ 10 and 15 kDa. Molecular weight marker (MW) in kDa is indicated to the left. Histones and histone variants names in *Plasmodium* species are indicated to the right. B: Immunoblots showing differential lysine lactylation of histones from synchronized trophozoites in response to 25 mM lactate in *P. knowlesi* and *P. falciparum* (biological triplicates). In each gel, lanes marked (+) and (-) correspond to ~ 1 ug of acid-extracted histones with or without lactate treatment. H4 is used as a loading control. Quantified signals are displayed to the right, as bar graphs. Bars show the mean band intensity from biological replicates, normalized to H4. Error bars show the standard

**Table 2. Primers used in ChIP-qPCR.**

| | Target gene name and ID | Name | Sequence |
|---|---|---|---|
| Significantly | Histone H4 (PY17X_0944400) | Fw | GAAAGGGAGGAGCAAAGAGACATAG |
| | | Rev | CCAGAGATACGTTTAACACCACCTC |
| regulated | Lactate dehydrogenase, LDH (PY17X_1345100) | Fw | AACCCAGTTGATTGCATGGC |
| | | Rev | CCTTGAACATCACCAGGGTTTAC |
| CDS | ATP-dependent 6-phosphofructokinase, PFK11 (PY17X_0922000) | Fw | GCAGGAATAGCTGTTGAAAATAATTTGG |
| | | Rev | ATTCTGATTGTTTATAAAATAATACCCGGG |
| | AP2 domain transcription factor, AP2-G (PY17X_1440000) | Fw | ATTGTGCCGAAACAGTACCG |
| | | Rev | AGCTAACCCGTTGCAATTGG |
| | isoleucine--tRNA ligase, Ile-tRNA (PY17X_1442500) | Fw | CGACGCTGGATATAAACGAGAATAATG |
| | | Rev | AACAAACCCATAGCCTCACAATATC |
| Significantly | Histone H4 (PY17X_0944400) | Fw | AAAGTGATGATAATGCATAAATTCGC |
| | | Rev | GAGTTTTGAGTCGATTTTATTTTAAATGGC |
| regulated | Lactate dehydrogenase, LDH (PY17X_1345100) | Fw | CATACTCAAATGTGTACGTGCA |
| | | Rev | AAATAAAAGCCCAATGACAAGAAAG |
| 5'UTR | ATP-dependent 6-phosphofructokinase, PFK11 (PY17X_0922000) | Fw | TTAGCGCTATAAACAAGTTACAGCA |
| | | Rev | GCACCTGTAAATAATATATCCAATATGTAC |
| | AP2 domain transcription factor, AP2-G (PY17X_1440000) | Fw | ATTGTGCCGAAACAGTACCG |
| | | Rev | AGCTAACCCGTTGCAATTGG |
| | isoleucine--tRNA ligase, Ile-tRNA (PY17X_1442500) | Fw | CTTCTTTAATTTGATGCGACATGAAATC |
| | | Rev | GATTGTACAGAAAAGGGGAGGG |
| Not significantly | PIR protein (PY17X_0900055) | Fw | AAGGAACGTGATTACCAATTCG |
| | | Rev | TGCCTCCGTTTCAAACACAG |
| regulated | amino acid transporter, AAT1 (PY17X_1129800) | Fw | CAATTCCATTAAATTTTATTGCGACCTATC |
| | | Rev | ATTTTCTTTGAAACATATCATCTCCTTGAG |
| CDS | microsomal signal peptidase, MSPase (PY17X_1420400) | Fw | TAAAAGAGGTGATGTTGTATTGCTAATATC |
| | | Rev | TACTATCAAAAGAATCCAGTTTATTATCGC |
| | apical polar ring protein, APR1 (PY17X_0909100) | Fw.1 | CAGGGAAAAGAAGATAAAGATGGGTTAGAC |
| | | Rev.1 | CATCAGATTCGGTCATATAATAATGTTCTTC |
| | apical polar ring protein, APR1 (PY17X_0909100) | Fw.2 | AATAAAAGCGATGAGGAATATGAAGACAAC |
| | | Rev.2 | TTTCGCTTCTACCTATGTTATATAATTCGC |
| | golgi protein 1, PRP2 (PY17X_1420000) | Fw | CGTGGAGTAAGTGGAAGATGG |
| | | Rev | TGTCCTTCATCTGCCAAATAGC |
| | mitochondrial import inner membrane translocase subunit, TIM10 (PY17X_0609600) | Fw | TAGAAGAATGCAAAATTCATGTTGGG |
| | | Rev | ATGAACACATCTATCAACACAACTC |
| | GPI-anchor transamidase, GPI8 (PY17X_0921600) | Fw | GATCAAAATGTTATCTTTTACGATTCAACC |
| | | Rev | GTAATAGCTTCGATCGTTGTATAAACTTTC |
| | parasite-infected erythrocyte surface protein, PIESP1 (PY17X_0411000) | Fw | AAGGTATGTTCATTGGATATGGATTTAATG |
| | | Rev | TTGCATCCAAACATGTATTTTCAAAATGAG |

error of the mean. T-tests were performed to check for significant changes between groups (0 mM versus 25 mM lactate), n = 3/group, *** indicates $P$-value ≤ 0.01.
(PDF)

**S4 Fig. The effect upon _P. falciparum_ cell-cycle progression of treatment with 25mM lactate for 12h.** A: _P. falciparum_ parasitaemia, blind-counted by microscopy, at each timepoint of the two timecourse experiments shown in Fig 2A.

Parasites were treated with 25mM lactate for 12h from the early trophozoite stage (or were untreated in the control); then in both conditions parasites were washed and regular counts were continued to the point of complete reinvasion in the subsequent cycle. The mean and range of counts from two independent timecourses are shown; n ≥ 20 parasites per time-point. At 24h, a slight delay in reinvasion was apparent in the two lactate-treated timecourses, but reinvasion rates recovered to control levels by 36h. B, C: Staging of parasites by microscopy (n ≥ 20 per timepoint) for the timecourses in (A). ET, early trophozoite; MT, mid trophozoite; LT, late trophozoite; S, schizont; R, ring. A slight delay in progression of mature stages (i.e., late trophozoites and schizonts) is apparent in the two lactate-treated timecourses (B) compard to the controls (C), as it is in (A), but homogenous populations of rings appeared in both timecourses, +/- 25mM lactate, by 36h. (PDF)

**S5 Fig. Distribution of lysine lactylation: peptide intensities.** A) Violin plot representing lactylated peptide intensities detected by mass spectrometry for samples with (Lactate) and without (Control) 25mM lactate treatment, with the middle horizontal line representing the median. The yellow shape shows the distribution of the data, and the grey dots represent the $\log_2$ transformed lactylation values for individual peptides. B) Venn diagram representing histone lysine (K) sites of *Plasmodium* that can be either uniquely lactylated (KLa) (dark pink), acetylated (Kac) (yellow), or lactylated and acetylated interchangeably (light pink). (PDF)

**S6 Fig. KLa profiles on histones are more inducible after lactate exposure than other PTMs.** A) Heatmap representing normalized intensities of significantly differential PTM sites (Z-score transformed) after 25mM lactate treatment: histone lysine lactylation (KLa), acetylation (Kac), trimethylation (Kme3) and dimethylation (Kme2). Hierarchical clustering was performed for each species, *P. falciparum* and *P. knowlesi,* independently to cluster the PTM site expression profiles with euclidean distance and ward.D clustering algorithm. The colored bar on the left represents the categories of histone PTM (lactyl/acetyl/methyl). B) Heatmap representing normalized intensities of the few histone Kac sites that were significantly differential, and that can be interchangeably – and significantly differentially – lactylated as well, in response to 25mM lactate in *P. falciparum*. (PDF)

**S7 Fig. KLa profiles in the acid-extracted proteome of *P. knowlesi* and *P. falciparum*.** A) Heatmap representing normalized peptide intensities of KLa sites on histone (red) and non-histone (black) proteins (Z-score transformed). Data from *P. falciparum* mixed-stage culture, +/-25mM lactate treatment. Hierarchical clustering was performed to cluster protein expression profiles with euclidean distance and ward.D clustering algorithm. B) Heatmap representing normalized peptide intensities of KLa sites on histone (red) and non-histone (black) proteins (Z-score transformed). Data from *P. knowlesi* mixed-stage culture, +/-25mM lactate treatment. Hierarchical clustering was performed to cluster protein expression profiles with euclidean distance and ward.D clustering algorithm. (PDF)

**S8 Fig. Comparison between the *Plasmodium* proteome and lactate-induced PTM landscape shows little correlation between PTMs and protein abundance.** A) Volcano plots representing differential proteome changes in *Plasmodium* trophozoites in response to 25mM lactate treatment. X-axes represent $\log_2$ fold changes (FC) calculated from normalized intensities, and y-axes represents $-\log_{10}$ transformed scaled adjusted *P*-values, and scaled *P*-values, for *P. falciparum and P. knowlesi*, respectively. Colored dots represent significantly changing proteins, based on their significance threshold indicated by *P*-value. Dashed black horizontal and vertical lines represent *P*-value and $\log_2$FC significance thresholds. B) Scatter plot representing correlation between significantly changing lactylated peptides and their protein abundance. $\log_2$FC of the lactylated peptide is plotted on the x-axis and the corresponding proteome $\log_2$FC is plotted on the y-axis. Pearson correlation coefficient (Pearson R) between both entities for each *Plasmodium* species is shown in red. Histone and non-histone proteins are in light orange and sky-blue, respectively. The linear regression fit is shown

by the green line (*x* and *y* reflect log$_2$FC on x-axes and y-axes, respectively). Only proteins with 2 unique razor+ peptides were included in the analysis.
(PDF)

**S9 Fig. GO enrichment analysis of biological processes related to PTMs that are influenced by lactate exposure.**
GO terms are shown for the proteins with significantly differential PTMs, and for proteins that changed significantly in the chromatin-associated proteome, after 25mM lactate treatment. Each GO term is represented with the genes associated with it as up- (triangle) or/and down- (circle) regulated. Shapes are color-coded based on the magnitude of the fold change. The size of the shape, represented by Z-score, indicates whether the term is likely to be decreased (negative value) or increased (positive value). For simplicity, K(me3) and K(me2) are presented as K(me). *P.f* and *P.k* represent Plasmodium *knowlesi* and *falciparum*, respectively. KLa.Sig and KLa represent GO terms enrichment for differentially expressed KLa proteins and lactylated acid-extracted proteome, respectively. Dashed vertical lines separate *P. knowlesi* and *P. falciparum* GO enrichment.
(PDF)

**S10 Fig. GO enrichment analysis of cellular components related to PTMs that are influenced by lactate exposure.**
GO terms are shown for the proteins with significantly differential PTMs, and for proteins that changed significantly in the chromatin-associated proteome, after 25mM lactate treatment. Each GO term is represented with the genes associated with it as up- (triangle) or/and down- (circle) regulated. Shapes are color-coded based on the magnitude of the fold change. The size of the shape, represented by Z-score, indicates whether the term is likely to be decreased (negative value) or increased (positive value). For simplicity, K(me3) and K(me2) are presented as K(me). *P.f* and *P.k* represent Plasmodium *knowlesi* and *falciparum*, respectively. KLa.Sig and KLa represent GO terms enrichment for differentially expressed KLa proteins and lactylated acid-extracted proteome, respectively. Dashed vertical lines separate *P. knowlesi* and *P. falciparum* GO enrichment.
(PDF)

**S11 Fig. GO enrichment analysis of molecular function related to PTMs that are influenced by lactate exposure.**
GO terms are shown for the proteins with significantly differential PTMs, and for proteins that changed significantly in the chromatin-associated proteome, after 25mM lactate treatment. Each GO term is represented with the genes associated with it as up- (triangle) or/and down- (circle) regulated. Shapes are color-coded based on the magnitude of the fold change. The size of the shape, represented by Z-score, indicates whether the term is likely to be decreased (negative value) or increased (positive value). For simplicity, K(me3) and K(me2) are presented as K(me). *P.f* and *P.k* represent Plasmodium *knowlesi* and *falciparum*, respectively. KLa.Sig and KLa represent GO terms enrichment for differentially expressed KLa proteins and lactylated acid-extracted proteome, respectively. Dashed vertical lines separate *P. knowlesi* and *P. falciparum* GO enrichment.
(PDF)

**S12 Fig. Differential gene expression and KLa chromatin enrichment profiles in response to hyperlactataemia.** A) Principal component analysis of RNA-Seq gene expression data representing normalized counts of 4983 and 4712 genes from three biological replicates (Rn) corresponding to *P. yoelii* (top panel) and *P. berghei* (bottom panel), respectively, at early (pink), and late (green) stages of mouse infection. B) Violin plot representing the distribution and number of significantly up- and down- regulated genes (*P*-value ≤0.05 (Top panel) or *P*-value ≤0.01 and |log2FC| ≥ 1 (lower panel) in *P. yoelii* 17XL (green) and *P. berghei* ANKA (purple) in late-stage versus early-stage infection. Violin width is irrespective of data range. C) KLa enrichment in the CDS of 5 differentially expressed genes (upper panel) and 9 control genes (lower panel) in *P. yoelii* after varying lactate exposure. Barplots show the percentage of chromatin input recovery (% IPed) for one region of each gene. D) KLa enrichment upstream of 5 differentially expressed genes, as in (C). Upper panel: Y-axis

represents the percentage of chromatin input recovery (% IPed) for each ChIPed region. One biological replicate for each of three mice (i.e., three lactate conditions). Dots represent individual genes, horizontal line represents median, box represents interquartile range, whiskers represent variability outside the interquartile range (quartile -/ + 1.5 * IQR). Multiple comparison of means was performed using ANOVA and Tukey HSD post hoc tests at p-value < 0.05 for between-group comparaison. ANOVA, F = 4.413; d.fsum = 2; p-value = 0.0366). *** p-value ≤ 0.001; ** p-value ≤ 0.01; *p-value ≤ 0.05; ns, not significant. Lower panel: barplot as in (C).
(PDF)

**S13 Fig. GO enrichment analysis of genes differentially expressed at late versus early stage of infection in *P.yoelii* 17XL and *P.berghei* ANKA.** A) Molecular function gene ontology (GO) enrichment analysis of DE genes in *P. berghei* and *P. yoelii.* Plots show the top 15 GO terms enriched in molecular function. B) Top 15 GO terms enriched in cellular components of DE genes in *P. berghei* and *P. yoelii*. C) Top 15 pathways terms enriched in DE genes in *P. berghei* and *P. yoelii*. For all plots the outer circle displays scatterplots of the expression levels (log2FC) for the genes (dots) in each GO term, whereas the inner circle is a bar plot where the height of the bar indicates the significance of the term (−log10 P-value), and color corresponds to the Z-score, which indicates if the term is likely to be decreased (negative value) or increased (positive value).
(PDF)

**S1 Table. *P. falciparum* trophozoite mass spectrometry and differential expression analysis.**
(XLSX)

**S2 Table. *P. knowlesi* trophozoite mass spectrometry and differential expression analysis.**
(XLSX)

**S3 Table. *P. falciparum* mixed stages mass spectrometry analysis.**
(XLSX)

**S4 Table. *P. knowlesi* mixed stages mass spectrometry analysis.**
(XLSX)

**S5 Table. Gene ontology of differentially expressed PTMs sites and acid-extracted proteome components in *P. falciparum* and *P. knowlesi* trophozoites.**
(XLSX)

**S6 Table. RNA-Seq analysis and GO enrichment of DE genes in *P. berghei* and *P. yoelii.***
(XLSX)

**S7 Table. Source data for all Figs: Tab A (Fig 1): Histone lysine lactylation in *P. falciparum* and *P. knowlesi*.** Uncropped western blot images from the n = 6 replicate experiments quantified in Fig 1, showing 'Pan Kla' and histone H4 K12 lactylation, as well as total histone H4 as control. **Tab B (Fig 2): Endogenous histone lysine lactylation in *P. falciparum*.** Uncropped western blot images from the n = 2 replicate experiments quantified in Fig 2, showing 'Pan Kla' in *P. falciparum* parasites grown from 1% parasitaemia without media change for 72h *in vitro*. **Tab C (Fig 3): Histone lysine acetylation in *P. falciparum*.** Uncropped western blot images from the n = 6 replicate experiments quantified in Fig 3, showing H3Kac and histone H4Kac, as well as total histone H4 as control. **Tab D (Fig 8): Lactate-associated changes of gene expression in *P. yoelii* (ChIP-qPCR in gene CDS).** Source data for experiments quantified in Fig 8, showing lactate concentrations in the host's blood (mM) and corresponding values for *P. yoelii* histone KLa enrichment within the coding sequences (CDS) of genes selected for ChIP-qPCR. These values represent KLa enrichment for gene groups whose expression was either regulated or not significantly regulated under increasing host lactate conditions. **Tab E (S1**

Fig): **Correlation between blood lactate and histone lysine lactylation in *P. berghei*.** Uncropped western blot images for the data presented in S1 Fig, showing 'Pan Kla' levels in *P. berghei* exposed to different host lactate concentrations. Total histone H4 is shown as loading control. **Tab F (S3 Fig): Histone lysine lactylation validation of mass spectrometry data in *P. knowlesi* and *P. falciparum*.** Uncropped fluorescent western blot images from the samples used for mass spectrometry, confirming histone quality and detecting 'Pan Kla' signals. Total histone H4 is shown as loading control. **Tab G (S12 Fig): Lactate-associated changes of gene expression in *P. yoelii* (ChIP-qPCR in 5'UTR).** Source data for experiments quantified in S12 Fig, showing lactate concentrations in the host's blood (mM) and corresponding values for *P. yoelii* histone KLa enrichment in the 5′ untranslated region (5′UTR) of genes selected for ChIP-qPCR. These values represent KLa enrichment for gene groups whose expression was either regulated or not significantly regulated under increasing host lactate conditions.
(XLSX)

## Acknowledgments

We acknowledge the Cambridge Centre for Proteomics for mass spectrometry; the staff of Imperial College Central Biomedical Services for support with animal experiments; Till Voss (Swiss TPH) for the HP1-HA parasite line; and Rachana Ramarao and Anders Jensen for early work on this project.

## Author contributions

**Conceptualization:** Catherine J. Merrick.

**Data curation:** Ibtissam Jabre, Nana Efua Andoh.

**Formal analysis:** Ibtissam Jabre, Nana Efua Andoh.

**Funding acquisition:** Catherine J. Merrick.

**Investigation:** Ibtissam Jabre, Nana Efua Andoh, Juliana Naldoni, William Gregory, Haddijatou Mbye, Chae Eun Yoon, Athina Georgiadou.

**Methodology:** Athina Georgiadou, Andrew M. Blagborough.

**Project administration:** Catherine J. Merrick.

**Resources:** Aubrey J. Cunnington, Athina Georgiadou, Andrew M. Blagborough.

**Supervision:** Aubrey J. Cunnington, Athina Georgiadou, Andrew M. Blagborough, Catherine J. Merrick.

**Visualization:** Ibtissam Jabre, Nana Efua Andoh, Juliana Naldoni.

**Writing – original draft:** Ibtissam Jabre, Nana Efua Andoh, Catherine J. Merrick.

**Writing – review & editing:** Ibtissam Jabre, Nana Efua Andoh, Catherine J. Merrick.

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
