## [Decision Letter · Decision Letter 0]

27 Jul 2025

PGENETICS-D-25-00592

Histone lactylation: a new epigenetic mark in the malaria parasite Plasmodium

PLOS Genetics

Dear Dr. Merrick,

Thank you for submitting your manuscript to PLOS Genetics. After careful consideration, we feel that it has merit but does not fully meet PLOS Genetics's publication criteria as it currently stands. Therefore, we invite you to submit a revised version of the manuscript that addresses the points raised during the review process.

Please submit your revised manuscript within 60 days Sep 25 2025 11:59PM. If you will need more time than this to complete your revisions, please reply to this message or contact the journal office at plosgenetics@plos.org. Please include the following items when submitting your revised manuscript:

We look forward to receiving your revised manuscript.

Kind regards,

Marnie E. Blewitt

Section Editor

PLOS Genetics

John Greally

Section Editor

PLOS Genetics

Aimée Dudley

Editor-in-Chief

PLOS Genetics

Anne Goriely

Editor-in-Chief

PLOS Genetics

**Journal Requirements:**

At this stage, the following Authors/Authors require contributions: Catherine J. Merrick. Please ensure that the full contributions of each author are acknowledged in the "Add/Edit/Remove Authors" section of our submission form.

The list of CRediT author contributions may be found here: https://journals.plos.org/plosgenetics/s/authorship#loc-author-contributions

2) We noticed that you used the phrase 'data not shown' in the manuscript. We do not allow these references, as the PLOS data access policy requires that all data be either published with the manuscript or made available in a publicly accessible database. Please amend the supplementary material to include the referenced data or remove the references.

4) We note that your Data Availability Statement is currently as follows: "Sequencing data from mouse, P.yoelii and P. berghei are available at ENA under the ID PRJEB43641 as per Georgiadou et al. [14]. Proteome data are deposited at the ProteomeXchange Consortium via the PRIDE partner with dataset identifier PXD055236.". Please confirm at this time whether or not your submission contains all raw data required to replicate the results of your study. Authors must share the “minimal data set” for their submission. PLOS defines the minimal data set to consist of the data required to replicate all study findings reported in the article, as well as related metadata and methods (https://journals.plos.org/plosone/s/data-availability#loc-minimal-data-set-definition).

2) If any authors received a salary from any of your funders, please state which authors and which funders..

6) Please send a completed 'Competing Interests' statement, including any COIs declared by your co-authors. If you have no competing interests to declare, please state "The authors have declared that no competing interests exist". Otherwise please declare all competing interests beginning with the statement "I have read the journal's policy and the authors of this manuscript have the following competing interests"

**Reviewers' comments:**

Reviewer's Responses to Questions

**Comments to the Authors:**

Reviewer #1: The topic of the paper is highly interesting, and the finding of lactylation of histones and other proteins in Plasmodium species is an important and new finding. However, the study is mostly descriptive, and the paper severely suffers from far too low numbers of replicate analyses, rendering statistically relevant comparisons fairly unreliable.

Major comments:

1. In most or all experiments with Western blot analyses, an insufficient number of biological replicates is shown. For instance, only two datapoints per condition is available in several figures (e.g. figure 2, figure 3, but also other figures). This is obviously not acceptable and the number of replicates needs to be upscaled. For instance, in figure 3E, absolutely nothing can be concluded when comparing the control vs the lactate condition. In general, for this type of analysis an absolute minimum of 5 to 6 datapoints per condition (and per time point) is required, in particular since it is clear that there is variability present (although with the number of data points provided, it is even hard to get an idea of the variability). This is the absolute minimum to obtain somewhat reliable statistical comparisions. Uncropped images of Western blots should be shown in supplementary data.

2. Furthermore, it needs to be explained more clearly how many biological replicates per experiment were performed. Was only one biological replicate analyzed per experiment? This is not sufficient.

3. The quantification of the data in most Western blot experiments are normalized to the controls. However, the way this is done is not OK, since all controls appear normalized to themselves (resulting in a value of one), which does not allow to evaluate variability between controls, and which results in incorrect statistical evaluation. Normalisation should be performed to the mean of several biological replicates of the control condition. Also the individual control datapoints should be normalized to the mean, allowing to visualize and evaluate their variability. Furthermore, why is this normalization performed in most experiments, but not in fig S4?

4. The specificity of the used antibodies need to be documented. Can these Abs fully differentiate between e.g. acetylation and lactylation on these plasmodial histones? This needs to be documented and relevant controls need to be provided. Are these antibodies raised against Plasmodium histones, or against which species? The mass spec data show that these modifications are specific between different Plasmodium species, so how to understand the specificity of these antibodies? This needs to be clarified.

5. What is the rationale to expose early trophozoites to lactate? It would be more logical to expose schizont stages, which is the sequestering stage and most likely exposed to the highest concentrations of lactate?

Minor comments:

Introduction: there are many more possible causes of hyperlactatemia besides parasite lactate production and anaerobic respiration in host tissues due to parasite sequestration. This should be briefly mentioned.

Fig 1D contains a strange outlier. More biological replicates are required.

Correlation analyses: p-values have to be indicated. Please also add units (or fold change?) to Y axes of correlation graphs.

The authors indicate that P. berghei ANKA does not cause hyperlactataemia, but figure S1 shows values up to 15 mM. Please clarify.

Lactylation appears still present 6 hours after lactate withdrawal in Pk but not in Pf, please discuss. Or is this the result of insufficient sample numbers? See comments above.

Line 225: when stating a trend, it is essential to show the p-value.

Fig 4: please clarify what is HA.

Line 444: Hyperlactatemia is very common in severe malaria patients. The authors are encouraged to also analyse the lactylation in patient samples, this would strongly improve the study.

The ChIP experiment in figure 8 is hard to understand and should be better explained. Also explain clearly which 5 genes, how were these chosen?

Why using different ways of lactatemia measurement for the two murine parasites, are these fully comparable? Are both specific for L-lactate?

Reviewer #2: In their manuscript “Histone lactylation: a new epigenetic mark in the malaria parasite Plasmodium”, Jabre et al. describe histone lactylation as an epigenetic mark in malaria parasites for the first time. The manuscript is generally of great interest and technically divided into two sections. Histone lactylation is first investigated using quantitative Western blotting. This involves investigating the dynamics of histone lactylation in blood stage parasites of P. falciparum and P. knowlesi upon addition of lactate and the decrease in histone marking upon withdrawal of lactate. Subsequently, the histones and non-histone proteins that are lactylated are identified and categorized using mass spectrometry. Furthermore, it is described that lysines that are lactylated are usually also acetylated, and it is suspected that acetyltransferases also transfer lactate to histones.

My comments are of a more general nature:

1. Introduction: The introduction does not read well. It seems very generic and incoherent. It should be revised to provide a logical introduction to the topic of the study.

2. The results section should briefly describe the respective method used. For example, in line 99ff, it is not clear that this is quantitative Western blotting.

3. The blood stages were treated with lactate for 12 hours (or 8 hours). Why wasn't the treatment time shorter?

4. It is generally unclear to me whether sufficient biological replicates were performed in the various experiments (especially in the mass spec experiments). This should be better described, or if the biological replicates are missing, they must be added.

5. The word "crashing" is very colloquial. If possible, it should be replaced.

6. It is described how the same lysines in the histones can be both lactylated and acetylated. If I understand correctly, both marks are present on the lysine at the same time. However, it is unclear to me what this looks like at the molecular level. This should be visualized.

7. Acetylation reversal occurs via HDACs. Which enzyme reverses lactylation?

8. To obtain more information about whether histone acetyltransferases are responsible for lactylation, knockout studies should be conducted, either using chemical HAT inhibitors or existing HAT knockouts. The use of HDAC inhibitors such as TSA would also be interesting. Overexpression studies could also be conducted for HATs.

9. Which enzyme lactylates the non-histone proteins?

10. Line 295: The text should describe which KLa sites were discovered, or at least mention the common KLA sites between P. falciparum and P. knowlesi.

11. It has been shown that both the genes of sexual commitment markers and the proteins themselves (AP2-G, MDV1) are lactylated. Therefore, it should be investigated whether the addition of lactate and histone lactylation has an influence on sexual commitment. Does lactate trigger or inhibit sexual commitment?

12. Lines 520ff: The five genes should be named.

Minor:

Line 267: It is HP1

Lines 375/376: A word appears to be missing at the end of the sentence (after "were").

Figure caption Fig. 8. It must be made clear here that previously published data sets from [14] were analyzed.

Reviewer #3: Here, Jabre et al. investigate the relatively unexplored epigenetic modification of lactylation and how it may impact the transcriptional programme of the malaria parasite as the infection progresses. Hyperlactatemia is a clinical manifestation during Plasmodium infection, and this study examines the novel concept that the parasite senses this metabolite and in a very direct way, by histone lactylation, with the implication that it may alter gene expression levels. Proteomics has also identified lactylation on non-histone proteins, suggesting a potential role for this modification beyond histones, as previously reported in higher organisms.

The study first shows an apparent increase in histone lactylation by supplementation of lactate in the media, as well as a correlation in vivo between lactylation and blood lactate levels. Dynamics of lactylation in both P. falciparum and P. knowlesi were examined using a pulse, showing relatively rapid, but transient, modification. Using proteomics, the authors then mapped the specific histones and residues that are modified in both P. falciparum and knowlesi, allowing comparison of species-specific differences. Finally, the authors aim to correlate expression of genes at late stages of infection when lactate levels are elevated, with their association with lactylated histones. Overall this study describes a new epigenetic mark with implications for gene expression during different phases of malaria infection, and would be of broad interest to the community.

The authors may want to consider the following points for revision.

Main points:

1) Figure 8C shows a crucial piece of data that could be explained in more depth, as it aims to provide the connection between histone lacytlation and alterations in gene expression. If I’ve understood correctly, the authors first analyzed previously published data (Georgiadou et al., 2022) on gene expression from an in vivo P. yoelii infection, deriving a gene set of differentially expressed genes upregulated in late-stage infection (and thus potentially associated with higher lactate levels). They then perform their own experiment using ChIP of lactyl-histone sites in three infected mice. However, the transition from analysis of the previous work to their own experiment (page 23, line 517) could be more clearly delineated, as this was not immediately clear. Similarly it was not immediately obvious that the three different lactate levels shown in the figure legend of Fig.8C were obtained by sampling from the infected mice. Were these harvested at different times (as with the Georgiadou et al. paper) or was this a fortuitous result that a range of lactate levels were obtained, and did that also correlate with parasitemia.

The text states that “ChIP was used to determine whether a selection of P. yoelii genes differentially expressed in hyperlactataemic hosts were also differentially marked with lactyl lysine.” However, this may give the impression that ChIP was performed on all deregulated genes, when it was conducted on five genes. It was not clear why these five genes were selected, nor how many were upregulated versus downregulated. Including ChIP analysis for a broader set of genes from both upregulated and downregulated categories would strengthen the conclusions.

2) For the histone lactylation measurements in Fig. 2, it wasn’t clear which of these were statistically significant? Was testing performed?

3) Given elevated parasitemia in P. falciparum was shown by the authors to yield higher levels of lactate in the media, it would be informative for the in vivo P. yoelii experiment shown in Fig. 1F to know if the higher lactyl-histone modifications also correlated with parasitemia, if this is known.

Minor points

4) In Fig.2C, the graph shows the parasite culture ‘crashing’ at only ~0.5% parasitemia – is this an error in labelling? It would be surprising if this occurred at such a low parasitemia.

5) Supp. Fig 2 – it is unclear whether B or C shows the lactate-treated experiment, and the figure legend does not describe panel C.

**Have all data underlying the figures and results presented in the manuscript been provided?**

Reviewer #1: Yes

Reviewer #2: Yes

Reviewer #3: Yes

PLOS authors have the option to publish the peer review history of their article (what does this mean? ). If published, this will include your full peer review and any attached files.

**Do you want your identity to be public for this peer review?** For information about this choice, including consent withdrawal, please see our Privacy Policy .

Reviewer #1: No

Reviewer #2: **Yes: ** Gabriele Pradel

Reviewer #3: No

**Figure resubmission:**
---

## [Decision Letter · Decision Letter 1]

12 Oct 2025

PGENETICS-D-25-00592R1

Histone lactylation: a new epigenetic mark in the malaria parasite Plasmodium

PLOS Genetics

Dear Dr. Merrick,

Thank you for submitting your manuscript to PLOS Genetics. After careful consideration, we feel that it has merit but does not fully meet PLOS Genetics's publication criteria as it currently stands.

There were major concerns raised by the reviewers that have not been dealt with in the revised manuscript. I am providing the opportunity for you to address these major concerns with further experiments, in a second round of major revisions. If you do not feel you can address these concerns, I understand if you would need to withdraw your manuscript. Therefore, we invite you to submit a revised version of the manuscript that fully addresses the points raised during the review process.

Please submit your revised manuscript within 60 days Dec 11 2025 11:59PM. If you will need more time than this to complete your revisions, please reply to this message or contact the journal office at plosgenetics@plos.org. Please include the following items when submitting your revised manuscript:

We look forward to receiving your revised manuscript.

Kind regards,

Marnie E. Blewitt

Section Editor

PLOS Genetics

John Greally

Section Editor

PLOS Genetics

Aimée Dudley

Editor-in-Chief

PLOS Genetics

Anne Goriely

Editor-in-Chief

PLOS Genetics

**Journal Requirements:**

1) We do not publish any copyright or trademark symbols that usually accompany proprietary names, eg ©,  ®, or TM  (e.g. next to drug or reagent names). Therefore please remove all instances of trademark/copyright symbols throughout the text, including:

- TM on page: 45.

2) We have noticed that you have uploaded Supporting Information files, but you have not included a complete list of legends. Please add a full list of legends for your Supporting Information file (Source data FINAL.xlsx)  after the references list.

3) Thank you for stating "Proteome data are deposited at the ProteomeXchange Consortium via the PRIDE partner with dataset identifier PXD055236. " We strongly recommend all authors decide on a data sharing plan before acceptance, as the process can be lengthy and hold up publication timelines. Please note that, though access restrictions are acceptable now, your entire data will need to be made freely accessible if your manuscript is accepted for publication. This policy applies to all data except where public deposition would breach compliance with the protocol approved by your research ethics board. 

4) Please ensure that the funders and grant numbers match between the Financial Disclosure field and the Funding Information tab in your submission form. Note that the funders must be provided in the same order in both places as well. Currently, "Sir Isaac Newton Trust, Alborada Fund and University of Cambridge JRG Scheme" are currently missing from the Funding Information tab.

**Reviewers' comments:**

Reviewer's Responses to Questions

Reviewer #1: Although some of the issues have been solved, two main problems in this paper remain. The lack of reasonable numbers of datapoints remains a major problem, as well as the way the Western blots have been normalized.

Major comments

1. The authors still do not provide a sufficient number of datapoints per condition. 2 / 3 /4 datapoints per conditions is not sufficient. Essentially the whole paper appears based on such low numbers of datapoints, this is not OK. The fact that Western blot is a semiquantitative methods is not a good reason to validate such low sample numbers, semiquantitative methods may rather require extra confirmation. Referring to other papers in which such low sample numbers might have been used is equally not a good rationale. I agree with the necessity to quantify Western blots by e.g. densitometry, and the main reason for such quantification is to apply statistics on a reasonable number of samples, enabling to draw solid conclusions. With such low sample numbers and considerable variation in many of the graphs, the conclusions are not sufficiently backed up by the data.

2. The problem of normalization of the Western blot data has not been solved. In particular, controls are normalized to themselves, resulting in the problem that all controls get the value of 1. This is statistically not OK, as it does not permit to evaluate the variation in the control condition. The authors refer in their reply to ‘The Design of a Quantitative Western Blot Experiment’ (2014) https://doi.org/10.1155/2014/361590). However, they do not apply at all the normalization method proposed in this paper, which suggests to normalize the loading controls to a (reference) control sample that is loaded onto lane 1 of all blots. Literally from this paper: “control is typically a pooled homogenate from all of the samples in a given study aliquoted into multiple tubes to permit the loading of a fresh control sample in lane 1 of each study blot”. In table 2 of that paper an example is given, in which a clear standard deviation (different from 0) is shown. Alternatively, 3 or 4 control samples must be loaded on each gel, and the data normalized to the mean of these controls.

Reviewer #2: I am satisfied with the responses to my comments and have no further comments.

Reviewer #3: The authors performed a number of textual modifications in response to the critiques. They did not choose to include additional genes in the ChIP analysis (Q3), however I’ll accept that their claims are largely restricted to those genes that were analysed.

For the question on replicates (Q2), which was also raised in one way or another by the other reviewers, the authors indicate that they are only showing qualitative trends (e.g. across timepoints). However, that’s not quite how the text reads.

For example, for Figure 2 (line 177-180): “We repeated these experiments in P. knowlesi (fig 2B): again, the histone lactylation signal was inducible across the trophozoite stage of the cell cycle, while background levels of lactylated histones were low, variable, and not clearly cell-cycle-regulated in this species (fig 2B: grey bars)”. This implies addition of lactate is inducing a quantitative increase in lactylated histones relative to the no-lactate condition. But, for example at timepoint 6/MT, one datapoint is the same as the no-lactate control and one is high.

Similarly for Figure 3E-H (line 246-248): “Across the normal cell cycle, fluctuations in acetylation were less than 2-fold, but after exposure to 25mM lactate, histones were, again,

additionally acetylated, particular on H4 in P. falciparum.” The claim that they are additionally acetylated relative to the control is a quantitative comparison, but the differences shown are small that it is hard to know if these are biologically meaningful.

In response to reviewer 1, the authors note that the original lactylation Nature paper quantified blots from two independent replicates (e.g. Fig3b,c). However, those were also done in technical triplicate, which should make the data more robust.

Thus it seems that either additional replicates would be required to substantiate the claims of lactate induction (potentially just at a single timepoint if not a whole timecourse) or a more cautious interpretation is required.

**Have all data underlying the figures and results presented in the manuscript been provided?**

Reviewer #1: Yes

Reviewer #2: Yes

Reviewer #3: Yes

PLOS authors have the option to publish the peer review history of their article (what does this mean? ). If published, this will include your full peer review and any attached files.

**Do you want your identity to be public for this peer review?** For information about this choice, including consent withdrawal, please see our Privacy Policy .

Reviewer #1: No

Reviewer #2: **Yes: ** Gabriele Pradel

Reviewer #3: No

**Figure resubmission:**
---

## [Editor Report · Decision Letter 2]

12 Dec 2025

Dear Dr Merrick,

We are pleased to inform you that your manuscript entitled "Histone lactylation: a new epigenetic mark in the malaria parasite Plasmodium" has been editorially accepted for publication in PLOS Genetics. Congratulations!

Yours sincerely,

John M. Greally, D.Med., Ph.D.

Section Editor

PLOS Genetics

John Greally

Section Editor

PLOS Genetics

Aimée Dudley

Editor-in-Chief

PLOS Genetics

Anne Goriely

Editor-in-Chief

PLOS Genetics

BlueSky: @plos.bsky.social

Comments from the reviewers (if applicable):

**Data Deposition**

http://datadryad.org/submit?journalID=pgenetics&manu=PGENETICS-D-25-00592R2

**Press Queries**

---

## [Editor Report · Acceptance letter]

PGENETICS-D-25-00592R2

Histone lactylation: a new epigenetic mark in the malaria parasite *Plasmodium*

Dear Dr Merrick,

We are pleased to inform you that your manuscript entitled "Histone lactylation: a new epigenetic mark in the malaria parasite *Plasmodium* " has been formally accepted for publication in PLOS Genetics! Your manuscript is now with our production department and you will be notified of the publication date in due course.

With kind regards,

Anita Estes

PLOS Genetics

On behalf of:
